# Anatomy of the energetic driving force for charge generation in organic solar cells

Kyohei Nakano [1], Yujiao Chen[1], Bo Xiao[2], Weining Han[3], Jianming Huang[1], Hiroyuki Yoshida[3,4], Erjun Zhou[2] & Keisuke Tajima [1]

Eliminating the excess energetic driving force in organic solar cells leads to a smaller energy loss and higher device performance; hence, it is vital to understand the relation between the interfacial energetics and the photoelectric conversion efficiency. In this study, we systematically investigate 16 combinations of four donor polymers and four acceptors in planar heterojunction. The charge generation efficiency and its electric field dependence correlate with the energy difference between the singlet excited state and the interfacial charge transfer state. The threshold energy difference is 0.2 to 0.3 eV, below which the efficiency starts dropping and the charge generation becomes electric field-dependent. In contrast, the charge generation efficiency does not correlate with the energy difference between the charge transfer and the charge-separated states, indicating that the binding of the charge pairs in the charge transfer state is not the determining factor for the charge generation.

[1] RIKEN Center for Emergent Matter Science (CEMS), 2-1 Hirosawa, Wako, Saitama 351-0198, Japan. [2] CAS Key Laboratory of Nanosystem and Hierarchical Fabrication, CAS Center for Excellence in Nanoscience, National Center for Nanoscience and Technology, 100190 Beijing, People's Republic of China. [3] Graduate School of Engineering, Chiba University, 1-33 Yayoi-cho, Inage-ku, Chiba-shi, Chiba 263-8522, Japan. [4] Molecular Chirality Research Center, Chiba University, 1-33 Yayoi-cho, Inage-ku, Chiba-shi, Chiba 263-8522, Japan. Correspondence and requests for materials should be addressed to E.Z. (email: zhouej@nanoctr.cn) or to K.T. (email: keisuke.tajima@riken.jp)

In a single-particle state picture, the photoelectric conversion process in organic solar cells (OSCs) involves the transition from an initial singlet ($S_1$) excited state with energy $E_g^{opt}$ generated by light absorption to a final charge-separated (CS) state with energy $E_{CS}$ (Fig. 1). In the CS state, there is a pair of free positive and negative charges, which are present in the donor and acceptor, respectively. Thus, $E_{CS}$ can be regarded as the energetic difference between the onset energy of the lowest unoccupied molecular orbital band ($E_{LUMO}^A$) for the acceptor and that of the highest occupied molecular orbital band ($E_{HOMO}^D$) for the donor. The enthalpy difference between the two states ($E_g^{opt} - E_{CS}$) is the primary energetic driving force for the overall charge generation. In high-performance OSCs, internal quantum efficiencies (IQEs) as high as 100%[1,2] and fill factors (FFs) of upto 80%[3,4] have been reported, comparable with inorganic and perovskite solar cells[5]. The high FF is associated with flat current-voltage curves around the short-circuit condition, which indicates that the charge generation efficiency is independent of the electric field at the donor/acceptor (D/A) interfaces. These observations have proved that efficient, electric field-independent photoelectric conversion using organic semiconductors is possible with a sufficiently large driving force of $E_g^{opt} - E_{CS}$. However, this excess energy ($E_g^{opt} - E_{CS}$) is wasted as heat, resulting in large overall energy loss in the form of low open-circuit voltage ($V_{OC}$). This fundamental trade-off between the energetic driving force for charge generation and the cell voltage is a reason for the power conversion efficiency (PCE) of OSCs being limited to 15% to date[6]. Therefore, the essential question in realizing efficient OSCs beyond the current limit is how much the excess energy can be reduced to increase $V_{OC}$ while maintaining efficient charge generation.

Several experimental studies have shown that the relaxed charge transfer (CT) state at the D/A interface is the main precursor for the charge separation process[7–9]. The overall energy loss during the charge generation can, therefore, be separated into two components: the exothermic transition from the $S_1$ to CT states and the subsequent endothermic transition from the CT to CS states, written as

$$E_g^{opt} - E_{CS} = \left(E_g^{opt} - E_{CT}\right) - (E_{CS} - E_{CT}), \quad (1)$$

where $E_{CT}$ is the lowest energy of the CT state. $E_g^{opt} - E_{CT}$ represents the energetic loss to form the CT state and $E_{CS} - E_{CT}$ corresponds to the binding energy of the charge pairs in the CT state.

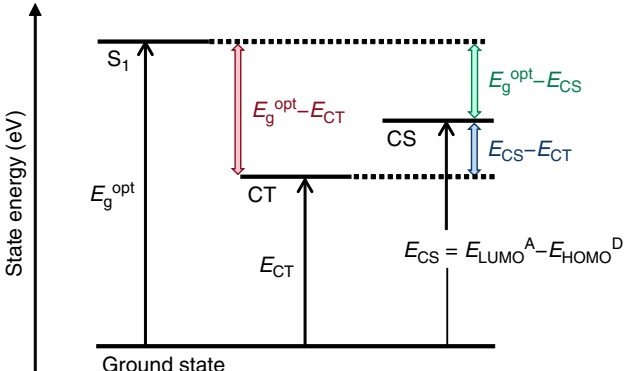

**Fig. 1** Schematic of the state energy. $E_g^{opt}$, $E_{CT}$, and $E_{CS}$ are the energy of singlet excited state, charge transfer state and charge-separated state, respectively. $E_{CS}$ was determined by the energetic difference between the onset energy of the lowest unoccupied molecular orbital band for the acceptor and that of the highest occupied molecular orbital band for the donor

Recently, $E_g^{opt} - E_{CT}$ has been used as an empirical indicator to show the low energy loss for mixed bulk heterojunction (BHJ) OSCs. High photoelectric conversion efficiency with $E_g^{opt} - E_{CT}$ of 0.1 eV or even smaller has been proposed[10–12]. However, there is a fundamental problem in estimating the interfacial energetics of BHJs due to their microscopic inhomogeneity; the disorder and the molecular intermixing can change the molecular properties at the D/A interface substantially compared with those in the bulk[13–15]. Therefore, for BHJs, using the bulk $E_g^{opt}$, $E_{HOMO}^D$, and $E_{LUMO}^A$ values measured for the pristine materials to calculate the overall energy loss is questionable. Moreover, large deviations in molecular properties near the D/A interface in BHJs make it difficult to discuss the data reported from different groups for various material combinations. In contrast, planar heterojunctions (PHJs) are a suitable structure for investigating the direct correlation between the interfacial properties and the device performance[16–23] because of their well-defined interfacial structure. Even with PHJs, however, reliable discussion of the interfacial energetics has suffered from the uncertainty of $E_{LUMO}^A$ due to the lack of accurate measurement methods, leading to a lack of quantitative studies of the intrinsic link between the interfacial energetics and the photoelectric conversion process.

In this study, we systematically investigate the charge generation in OSCs with 16 combinations of four donor polymers and four acceptors using PHJ structure. A well-defined interface of PHJs allows us to eliminate the effects of the complicated mixed interfaces and the inhomogeneity of the structures. We also use reliable electronic PHJ properties, such as $E_{HOMO}^D$ and $E_{LUMO}^A$, which are directly measured by ultraviolet photoemission spectroscopy (UPS) and low-energy inverse photoemission spectroscopy (LEIPS)[24,25] for each pristine film. The correlations of the charge generation efficiency and its electric field dependence with the energetic difference of $E_g^{opt} - E_{CT}$ and $E_{CS} - E_{CT}$ are investigated to explore the minimum requirement of the energetic driving force in efficient OSCs.

## Results

**Molecular structures and energy levels of materials.** Figure 2a shows the molecular structures of the materials used in this study. P3HT, PDCBT, PTB7, and J61 were used as representative polymer donors. These polymers have different electronic structures, although they can all show high external quantum efficiencies (EQEs) of more than 65% in mixed BHJ structures with the proper acceptors[26–29]. One fullerene derivative (PCBM) and three non-fullerene materials (BTAs) were used as the acceptors. BTAs have similar core π-conjugated structures but different energy levels due to their different end functional groups. Using BTAs as the acceptor in BHJ-type OSCs gave a high $V_{OC}$ (1.15–1.30 eV) with a PCE of upto 8.25%[30–32].

Figure 2b shows the energy levels of the eight materials. $E_{HOMO}^D$ and $E_{LUMO}^A$ values were measured by UPS (Supplementary Fig. 1) and LEIPS (Supplementary Fig. 2), respectively. The energy levels are presented with the aligned Fermi level ($E_F = 0$) because of the equilibrated charge distributions when the materials are in contact. The energy levels with respect to the vacuum level are shown in Supplementary Fig. 3. The cross point of the optical absorption and emission spectra for the film state was used to determine the energy of the $S_1$ state ($E_g^{opt}$; Supplementary Fig. 4). The dotted lines in Fig. 2b indicate $E_{HOMO}^D + E_g^{opt}$ and $E_{LUMO}^A - E_g^{opt}$ for the donors and the acceptors, respectively. $E_{CS}$ of the PHJ systems was defined as the difference between $E_{LUMO}^A$ and $E_{HOMO}^D$ for each combination of materials (Supplementary Table 1).

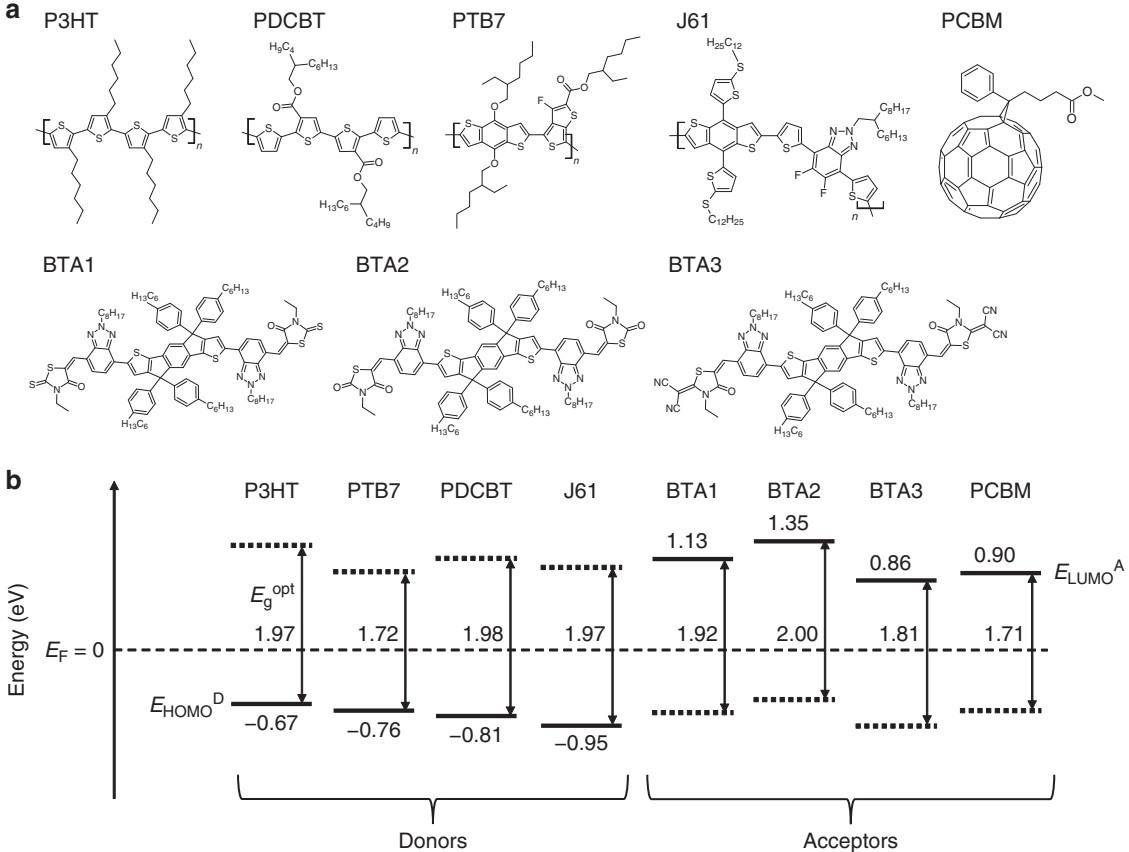

**Fig. 2** Materials and energy levels. **a** Molecular structures of the materials used in this study. **b** Schematics of the energy levels of the materials with respect to the aligned Fermi level ($E_F = 0$). Solid lines indicate the energy of the HOMO onset for the donor ($E_{HOMO}^D$) and LUMO onset for the acceptor ($E_{LUMO}^A$), evaluated by UPS and LEIPS, respectively. $E_g^{opt}$ are optical band gaps determined by absorption and emission spectra. The dotted lines indicate $E_{HOMO}^D + E_g^{opt}$ and $E_{LUMO}^A - E_g^{opt}$ for the donors and the acceptors, respectively

**Solar cell characteristics**. The PHJ devices with the combinations of the four donor polymers and the four acceptors were prepared. Supplementary Table 2 shows the film preparation conditions. Each donor film was transferred onto the acceptor film by the water-assisted contact film transfer method reported elsewhere[33,34] (see also the Methods section for details). The device structure was indium tin oxide (ITO)/polyethylenimine ethoxylated (PEIE)/acceptor/donor/MoO$_x$/Ag. Current density-voltage ($J$-$V$) characteristics were recorded under AM1.5 100 mW cm$^{-2}$ simulated sunlight irradiation. The $J$-$V$ characteristics and EQEs are shown in Supplementary Figs. 5 and 6. Short circuit current density ($J_{SC}$), $V_{OC}$, FF, and PCE of all the OSCs are summarized in Supplementary Table 3.

**Evaluation of $E_{CT}$**. Two main empirical methods have been proposed to obtain $E_{CT}$ at the D/A interface of OSCs. First, we measured the temperature ($T$) dependence of the $J$-$V$ characteristics under light irradiation and evaluated $E_{CT}$ by linearly extrapolating the $qV_{OC}$-$T$ plot to 0 K according to the literature[35,36]. This method relies on the assumptions that the non-geminate charge recombination occurs exclusively through the single manifold of the interfacial CT state and that the non-radiative component of the charge recombination does not occur at 0 K. We observed that the $qV_{OC}$-$T$ plots for all the combinations of materials showed linear relationships in the range of 300 to 210 K (Supplementary Fig. 7), as previously reported[35,36]. We used the intersections of $qV_{OC}$ at 0 K as $E_{CT}$ in this study, although it has been suggested that there may be differences of

less than 0.2 eV between $qV_{OC}$ at 0 K and $E_{CT}$ at room temperature due to the temperature dependence of $E_{CT}$[35].

We also conducted highly sensitive EQE and electroluminescence (EL) measurements to evaluate the absorption and emission of CT states for all the systems (Supplementary Fig. 8). Fitting the edges of the EQE and EL spectra with Gaussian functions with common parameters gives $E_{CT}$ as the cross point of the two Gaussians and the reorganization energy, $\lambda$, as the width of the functions according to the formula based on Marcus electron transfer theory[35]. For four of the 16 PHJ systems, we observed well-resolved CT bands and obtained $E_{CT}$ values using the optical measurements with reasonable fittings. However, six systems had lower goodness of fit due to deformations in the peak shapes, and a further six systems had $E_{CT}$ bands that were poorly separated from the $S_1$ bands and it was impossible to extract information about the CT states. The extracted values are summarized in Supplementary Table 1.

The $E_{CT}$ values determined by $qV_{OC}$-$T$ plots and EQE/EL measurements matched well for eight systems, with deviations below 15% (Supplementary Fig. 9 and Supplementary Table 1), whereas two PHJ systems with a low goodness of fit showed larger deviations (47 and 30% for BTA3/P3HT and PCBM/P3HT, respectively). The accuracy of the EQE/EL technique may be limited in PHJs because the small D/A interface area led to a smaller interfacial absorption/emission signal relative to that of the bulk, which reduced the reliability of the Gaussian fittings. Because of the lower reliability of $E_{CT}$ determined by EQE/EL measurements, we decided to focus on the $E_{CT}$ values from the $qV_{OC}$-$T$ plots. However, using the $E_{CT}$ values obtained by the

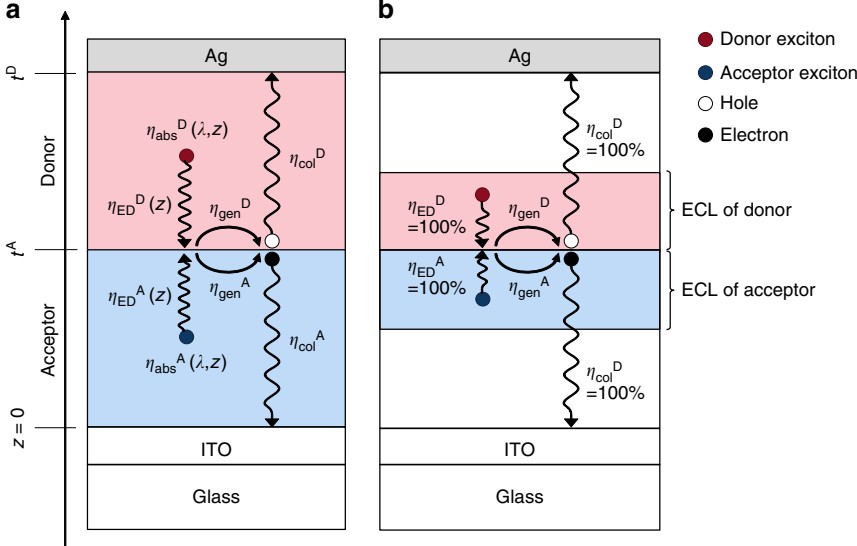

**Fig. 3** Schematic illustrations of the charge generation process. **a** Schematic of the processes with their efficiencies ($\eta$), where the superscripts D and A indicate that the process occurs in the donor and the acceptor, respectively. Upon light irradiation, the excitons are generated with efficiency $\eta_{abs}(\lambda, z)$ and diffuse to reach the D/A interface with efficiency $\eta_{ED}(z)$. The excitons are converted into free charges with efficiency $\eta_{gen}$ and collected in each electrode with efficiency $\eta_{col}$. **b** Schematic of the assumptions in this study used to reproduce the experimental external quantum efficiency spectra. All the excitons generated around the D/A interface within the exciton collection length (ECL) reach the interface ($\eta_{ED} = 100\%$ within ECL and $\eta_{ED} = 0\%$ outside ECL) and all the generated charges are collected by the electrodes ($\eta_{col} = 100\%$) (see Supplementary Fig. 11)

EQE/EL measurements did not change our discussion and conclusions.

**Evaluation of interfacial charge generation efficiency.** In OSCs based on PHJs, the EQE at wavelength $\lambda$ can be expressed as

$$\mathrm{EQE}(\lambda) = \int_0^{t^A} \eta_{abs}^A(\lambda, z)\eta_{ED}^A(z)\eta_{gen}^A\eta_{col}^A \mathrm{d}z + \int_{t^A}^{t^D} \eta_{abs}^D(\lambda, z)\eta_{ED}^D(z)\eta_{gen}^D\eta_{col}^D \mathrm{d}z,$$

(2)

where $t$ is the layer thickness, $\eta_{abs}(\lambda, z)$ is the light absorption efficiency at position $z$ (with the origin at ITO/acceptor interface), $\eta_{ED}(z)$ is the efficiency for the excitons generated at position $z$ to diffuse to the D/A interface, $\eta_{gen}$ is the interfacial charge generation efficiency from the exciton, and $\eta_{col}$ is the charge collection efficiency (Fig. 3a). Note that the efficiencies can depend on whether the events happen at the donor or acceptor sides[37], which is indicated by the superscripts D and A, respectively.

In PHJs, non-geminate recombination has a negligible effect under short-circuit conditions, as shown in the transient photocurrent measurements with variable light intensity (Supplementary Fig. 10 and Supplementary Note 1). Therefore, $\eta_{col}$ under short-circuit conditions can be assumed as unity. This assumption is not necessarily true for BHJs and can be characteristic of PHJs because the electrons and holes are spatially separated in the thin acceptor and the donor layers, respectively.

To evaluate $\eta_{abs}(\lambda, z)$, we calculated light absorption profiles in the multilayered films by using the optical transfer matrix formalism following a reported procedure[38,39]. The optical constants of the eight materials and the other layers were evaluated separately by using a spectroscopic ellipsometer (see Methods and Supplementary Note 2 for details). By using these optical constants and the films thicknesses determined by X-ray reflectivity, the simulated reflectance spectra of the OSC devices with normal incident light reproduced the experimental spectra well for all the PHJ systems (Supplementary Fig. 11), which supports the validity of the optical models. These analyses give the full energy dissipation profiles, $\eta_{abs}(\lambda, z)$, which indicate how

much light at each wavelength is absorbed at each position in the multilayered films.

To reproduce the EQE spectra with $\eta_{gen}$ as the unknown parameter, in principle, the diffusion equation can be solved numerically in PHJs to obtain $\eta_{ED}$ from the diffusion constants and the exciton lifetimes. However, accurate evaluation of these parameters is difficult due to many factors, such as the dimensionality of the diffusion process. Indeed, the reported diffusion constants and diffusion length vary greatly in the literature depending on the measurement method[40]. Therefore, we used a simple model assuming the exciton collection length (ECL) near the D/A interface for each material and that all the generated excitons reach the D/A interface (Fig. 3b). The ECLs of each material were determined to reproduce the observed EQE spectra best for the PHJ systems. Note that the lower limits of ECL were determined by the restriction that the maximum $\eta_{gen}$ is 100% and we allowed practical variations of ECL (13 to 18 nm for PCBM and 7 to 12 nm for other materials) to estimate the possible error range of $\eta_{gen}$. The uncertainty of the ECL is shown as the error bars for calculated $\eta_{gen}$. The EQE spectra calculated with these ECLs and $\eta_{gen}$ reproduced the experimental EQE spectra of most of the systems well (see Supplementary Note 3 and Supplementary Fig. 11).

**Charge generation efficiency and state energy difference.** The logarithms of $\eta_{gen}$ were plotted against $E_g^{opt} - E_{CS}$, which is the overall energetic driving force of charge generation (Fig. 4a). The filled and open symbols show the charge generation from the excitons generated at the donor ($\eta_{gen}^D$) and the acceptor sides ($\eta_{gen}^A$), respectively. The error bars indicate the variations of the assumed ECL. There was a weak trend with scattering in which a smaller driving force produced lower $\eta_{gen}$. Interestingly, when the logarithms of $\eta_{gen}$ were plotted against $E_g^{opt} - E_{CT}$, the trend was much clearer with a threshold behavior (Fig. 4b). A strong positive correlation of $\eta_{gen}$ with $E_g^{opt} - E_{CT}$ was observed for systems where $E_g^{opt} - E_{CT}$ was smaller than 0.2 eV, whereas $\eta_{gen}$ reached a maximum in systems with $E_g^{opt} - E_{CT}$ larger than 0.2 eV. Then $\eta_{gen}$ decreased in systems with $E_g^{opt} - E_{CT}$ larger than

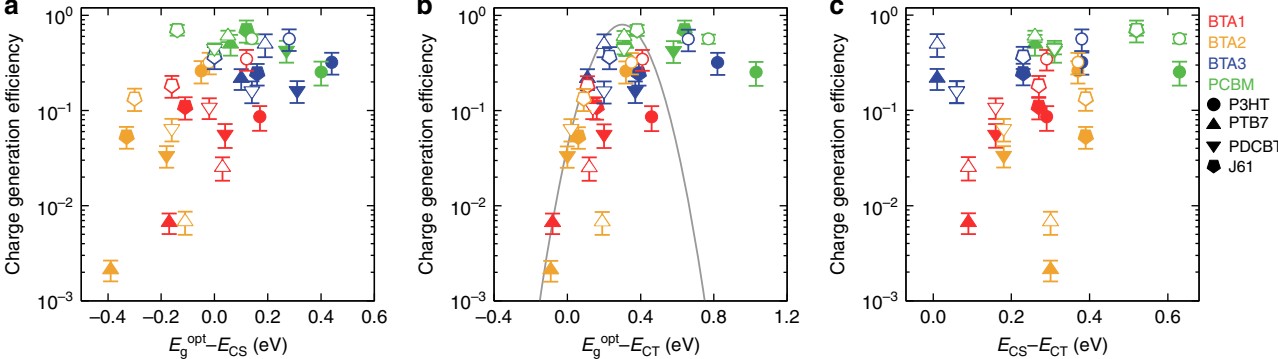

**Fig. 4** Charge generation efficiency and state energy difference. $\eta_{gen}$ plotted against **a**, $E_g^{opt} - E_{CS}$, **b**, $E_g^{opt} - E_{CT}$, and **c**, $E_{CS} - E_{CT}$. The filled and open symbols show the charge generation from the excitons generated at the donor and the acceptor sides, respectively. The error bar indicates the variation of the assumed exciton collection length. The gray line in panel **b** is a Gaussian derived from Marcus charge transfer theory with a reorganization energy of 0.3 eV

0.8 eV. This behavior is probably explained by the inverted region in Marcus charge transfer theory; the efficiency of the intermolecular charge transfer is at a maximum when the free energy difference between the initial and the final states is equal to the reorganization energy of the charge transfer event[41]. Similar behavior has been reported using time-resolved microwave conductivity[42] and photoluminescence (PL) quenching[43]. The clear correlation between $\eta_{gen}$ and $E_g^{opt} - E_{CT}$ suggests that the charge transfer process from the $S_1$ to CT states can be described by Marcus theory and that this process has a large effect on the overall charge generation process of organic solar cells. When we used the $E_{CT}$ values determined by EQE/EL measurements, the trend in the dependence of $\eta_{gen}$ on $E_g^{opt} - E_{CT}$ discussed above remained unchanged (Supplementary Fig. 12). We also checked the possible energy transfer between the donor and acceptor had little effect on the trend (See Supplementary Fig. 13 and Supplementary Note 4).

Note that the dispersion of $\eta_{gen}$ are too large to extract any meaningful information from the theoretical fit of Marcus theory to Fig. 4b. The charge generation process is probably a more complicated process than a single Marcus type transition and other contributions such as triplet state formation[44,45] should be considered for more detailed analysis. Also, the variations of the reorganization energy of each material[46] causes the dispersion. Instead of a theoretical fit, we plotted a Gaussian derived from Marcus theory with a reorganization energy of 0.3 eV in Fig. 4b (gray line; see detail in Supplementary Note 5). The upper limit of the charge generation efficiency well follows the Gaussian line in the Marcus normal region, supporting the primary role of the Marcus theory for the charge generation process of OSCs.

The charge transfer from the CT to CS state is endothermic due to the Coulombic binding of the charge pairs; hence, $\eta_{gen}$ was expected to increase as $E_{CS} - E_{CT}$ decreased. However, no clear correlation was observed between $\eta_{gen}$ and $E_{CS} - E_{CT}$ (Fig. 4c). This indicates that the effect of the CT binding energy ($E_{CS} - E_{CT}$) is not apparent on the charge generation of OSCs. In addition, there were large differences in $\eta^D_{gen}$ and $\eta^A_{gen}$ for each PHJ (filled and open symbols, respectively, with same shapes and colors in Fig. 4). This observation also implies the small effect of the CT binding energy on the charge generation process because the CT binding energy is independent of the origin of the excitation. This result is consistent with the previous report that the transition from the CT to CS states in the PTB7:PCBM BHJ system was a nearly barrier-free, efficient process[47]. The exact mechanism of this efficient CT state splitting is still unclear. An explanation has been proposed based on a kinetic Monte Carlo simulation that

the Coulombic binding of the CT states can be overcome by the relaxation of electrons (holes) in the disordered density of states of the acceptor (donor)[48]. It has also been proposed that the entropic gain plays a significant role in the CT states splitting into CS states[49,50]. Either way, we conclude that the energetic (enthalpic) driving force is important for the $S_1$ to CT transition, whereas factors other than the energetics are more important for CT states splitting into CS states.

**Electric field dependence of charge separation.** Next, we investigated the electric field dependence of the charge generation. For quantitative discussion, we defined the degree of the electric field dependence as the ratio of the charge generation efficiency with an applied bias of 0 V (*i.e.*, short circuit) to −1.0 V as

$$\text{Electric field dependence} = \left(1 - \frac{\eta_{gen}(0\,\text{V})}{\eta_{gen}(-1.0\,\text{V})}\right) \times 100\,(\%)$$

(3)

To evaluate $\eta_{gen}$ at a bias of −1.0 V, we conducted EQE measurements at a −1.0 V bias and reproduced the spectra by using the simulated light absorptance in the same way as for $\eta_{gen}$ under short-circuit conditions. Figure 5a, b show the electric field dependence plotted against $E_g^{opt} - E_{CT}$ and $E_{CS} - E_{CT}$, respectively. No clear trend was observed in the $E_{CS} - E_{CT}$ plot. In contrast, the charge generation become less dependent on the electric field at $E_g^{opt} - E_{CT}$ of around 0.4 eV.

The FF of the PHJs also decreased with $E_g^{opt} - E_{CT}$ below about 0.3 eV (Supplementary Fig. 14), whereas a similar but weaker trend was observed in the relationship with $E_{CS} - E_{CT}$. The former observation may also suggest that $E_g^{opt} - E_{CT}$ has a large effect on the electric field dependence of the charge generation. However, both geminate and non-geminate recombination can affect the FF and their complete separation would be difficult under the OSC operating conditions.

**Electric field dependence of emission from the $S_1$ state.** The clear correlations of $\eta_{gen}$, FF, and the electric field dependence of $\eta_{gen}$ with $E_g^{opt} - E_{CT}$ suggest that the transition from the $S_1$ to the CT states has a large effect on the overall charge generation process in OSCs. If the decay of the $S_1$ to the ground state competing with transition of the $S_1$ to CT state is the origin of the electric field dependence in the cells, the $S_1$ emission may also show electric field dependence. To test this, we selected four PHJ cells with different electric field dependences of $\eta_{gen}$ and

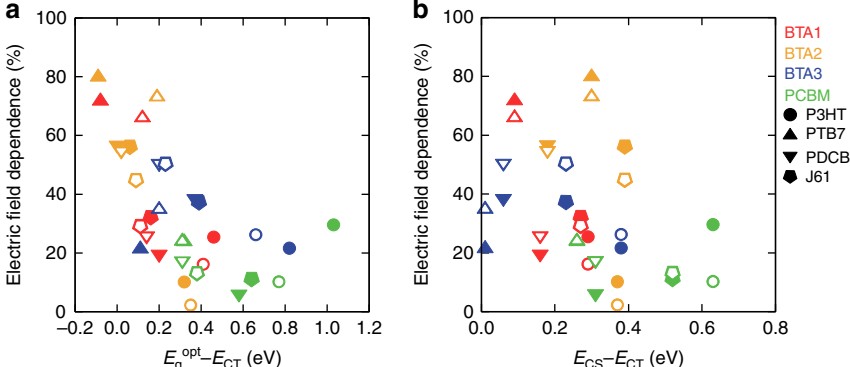

**Fig. 5** Electric field dependence of charge generation. The electric field dependence is defined as the ratio of the charge generation efficiency with an applied bias of 0 V (i.e., short circuit) to −1.0 V. Electric field dependence plotted against **a**, $E_g^{opt} − E_{CT}$ and **b**, $E_{CS} − E_{CT}$

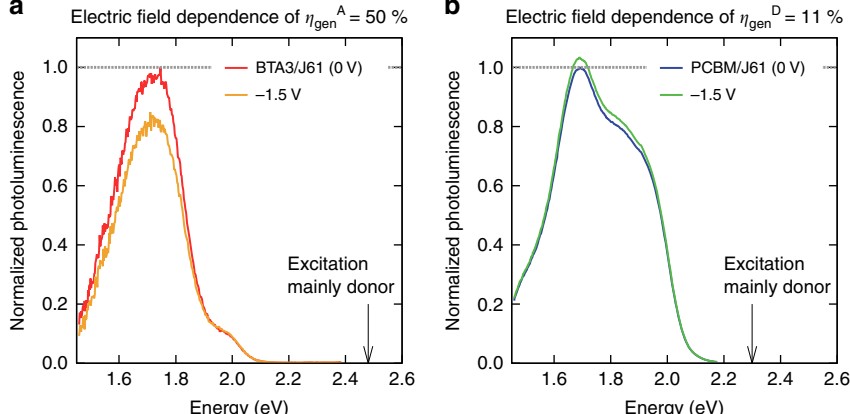

**Fig. 6** Normalized photoluminescence under different bias conditions. **a** BTA3/J61 with electric field dependence of $\eta_{gen}^A$ of 50%. The excitation wavelength of 500 nm is indicated by the arrow. **b** PCBM/J61 with electric field dependence of $\eta_{gen}^D$ of 11%. The excitation wavelength of 540 nm is indicated by the arrow. The spectra are normalized by the spectrum measured under an applied bias of 0 V. The decrease of the PL intensity for BTA3/J61 (**a**) with −1.5 V bias indicates that emission from the $S_1$ state decreased with increasing internal electric field. The PL spectra of the pristine films are shown in Supplementary Fig. 15

measured the PL with an applied reverse bias. Figure 6a, b compare the PL of BTA3/J61 and PCBM/J61 PHJ devices, respectively, under applied biases of 0 V and −1.5 V. The PL of the PHJs and pristine films is shown in Supplementary Fig. 15. The PL of PCBM/J61 upon excitation of J61 at 540 nm was assigned to the emission from the $S_1$ state of J61. In contrast, the PL of BTA3/J61 upon excitation of J61 at 500 nm was assigned to the emission from the $S_1$ state of BTA3. This is attributed either to the energy transfer from J61 to BTA3 prior to the emission or to the regeneration of the excited state of BTA3 through the CT state. We could not distinguish these processes, but the possible contribution of the energy transfer does not affect the following discussion because the efficiency of the energy transfer would not be affected by the electric field.

In the PCBM/J61 PHJ, the PL intensity was almost unchanged under a reverse bias of −1.5 V. This is consistent with the observation that the electric field dependence of $\eta_{gen}^D$ was small (11%) in this system. The $S_1$ state of J61 generated within the ECL was efficiently converted to the CT states regardless of the internal electric field. The $S_1$ state generated outside of the ECL (far from the D/A interface) decayed to the ground state, and the emission was mainly detected as PL. In contrast, in BTA3/J61 the PL from the $S_1$ state of BTA3 decreased substantially with an applied bias of −1.5 V. This is consistent with the relatively large field dependence of $\eta_{gen}^A$ (50%) in this system; the larger internal

electric field generated by the applied bias promoted the CT state formation and decreased the population of the $S_1$ state in BTA3, which decreased the PL signal intensity. We conducted the same experiments for BTA2/J61 (electric field dependences of $\eta_{gen}^A$ and $\eta_{gen}^D$ were 45% and 56%, respectively), PCBM/PTB7 (electric field dependence of $\eta_{gen}^D$ was 24%), and BTA2/P3HT (electric field dependence of $\eta_{gen}^D$ was 10%) and found that the electric field dependence of the $S_1$ emission appeared in only the BTA2/J61 PHJ (Supplementary Fig. 16). Based on these results, we conclude that the origin of the electric field dependence of the charge generation observed for small $E_g^{opt} − E_{CT}$ is the competition between the decay from the $S_1$ to the ground state and the transition from $S_1$ to CT state. The $E_g^{opt} − E_{CT}$ less than 0.2 eV reduces the transition rate from the $S_1$ to CT states[51] and increases the rate from the CT to $S_1$ states[52]. The electric field dependence of $S_1$ state dissociation has been reported using a BHJ device with a low driving energy[53,54], indicating that this observation is general, regardless of the device architecture.

## Discussion

Here we discuss the efficient BHJ OSCs with small energy driving force. There are several reports of efficient BHJ solar cells with a small $E_g^{opt} − E_{CT}$ (less than 0.1 eV)[10–12]. For these efficient cells, the charge generation does not depend on the electric field and the FFs are larger than 0.6. In this study, however, we observed a

stricter requirement, that $E_g^{opt} - E_{CT}$ larger than 0.2 to 0.3 eV is necessary for efficient electric field-independent charge generation. To explain this discrepancy, we hypothesize that $E_g^{opt} - E_{CT}$ of BHJ is underestimated. In the random mixture of the donor and acceptor material, the aggregation and crystallinity near the D/A interface differ from those of the bulk domain[13–15] because of the interfacial disorder or molecular intermixing[55,56]. In this situation, $E_g^{opt}$ at the interface becomes larger than that of the bulk due to the disorder, and $E_{CT}$ also increases due to a deeper $E_{HOMO}^D$ in the disordered donor or a shallower $E_{LUMO}^A$ in the disordered acceptor (Supplementary Fig. 17). However, the broadened optical gap in the disordered layer would be difficult to detect in the absorption spectra because its absorptance is small and the blue-shifted absorptance overlaps with the bulk absorptance. Hence, $E_g^{opt}$ of the pristine materials or apparent $E_g^{opt}$ is generally used for calculating $E_g^{opt} - E_{CT}$ of BHJs, which could cause the underestimation of the energy difference. Indeed, we showed previously that when a disordered polymer layer approximately 4 nm thick was intentionally placed between the donor and acceptor layers of a PHJ, the interfacial $E_{CT}$ value increased by approximately 0.25 eV without altering the $E_g^{opt}$ of the bulk[57]. A similar situation may occur unintentionally at the D/A interfaces in mixed BHJs. The discrepancy in the criteria for PHJs and BHJs highlights the importance of controlling interfacial structures, which could change the design principles of the organic semiconducting materials beyond simple energy level matching.

Finally, we discuss the energy loss of $E_g^{opt} - qV_{OC}$. Energy $qV_{OC}$ is equivalent to the splitting of the quasi-Fermi levels of holes and electrons under photoirradiation, which are related to the energy levels and the charge density at the steady state. Therefore, $E_g^{opt} - qV_{OC}$ includes both single-particle state energetics and the charge recombination kinetics. This kinetic loss can be described as the energy difference, $E_{CT} - qV_{OC}$. The total energy loss $E_g^{opt} - qV_{OC}$ is the sum of the energetics and kinetic losses[58],

$$E_g^{opt} - qV_{OC} = \left( E_g^{opt} - E_{CT} \right) + (E_{CT} - qV_{OC}) \qquad (4)$$

Figure 7a shows $E_{CT} - qV_{OC}$ of the PHJ OSCs plotted against $E_{CT}$. The loss of $E_{CT} - qV_{OC}$ is distributed around 0.5 eV with no clear correlation with $E_{CT}$. This suggests that the recombination kinetics depend weakly on the interfacial energetics. Benduhn et al. observed a slight decrease in this energy loss with increasing $E_{CT}$ in BHJs. They explained the dependence by using the energy gap law of non-radiative recombination[59], in which the electron transfer rate increases as the energy difference between the initial and final states decreases. However, we did not see this trend in our data set: 16 data points may be not enough to see such a subtle difference. The minimum threshold of $E_g^{opt} - E_{CT}$ is 0.2 to 0.3 eV for the charge generation and $E_{CT} - qV_{OC}$ is distributed around 0.5 eV, which accounts for the minimum $E_g^{opt} - qV_{OC}$ of 0.7 to 0.8 eV for the efficient OSCs. Indeed, $\eta_{gen}$ decreased dramatically when the overall energy loss was below this threshold (Fig. 7b). This result is roughly consistent with the previously observed trend that the EQE of the BHJ OSCs drastically decreases when the $E_g^{opt} - qV_{OC}$ loss of the system is below 0.6 eV[60].

However, there is a notable exception from the trend: the BTA3/P3HT system shows a significantly smaller $E_{CT} - qV_{OC}$ of 0.31 eV compared with other systems (Fig. 7a, blue filled circles), whereas the other combinations containing either BTA3 or P3HT followed the general trend. This means that the recombination loss relates to the properties of the D/A interface, especially the transition probability from the CT state to the ground state, through many factors, such as the electronic coupling, the effective molecular distance, and the molecular orientations between the donor and acceptor molecules. In other words, a low recombination loss may not be achieved by fine tuning the properties of each material, such as the energetics, but by careful optimization of the interfacial structures in the specific D/A combinations.

In 16 organic PHJ systems, the charge generation efficiency and the electric field dependence of the charge generation clearly correlated with the energy offset, $E_g^{opt} - E_{CT}$, but was not correlated with the binding energy of the CT state, $E_{CS} - E_{CT}$. The threshold energy required for the efficient charge generation was $E_g^{opt} - E_{CT}$ of 0.2 to 0.3 eV. Below the threshold, the $S_1$ state decayed to the ground state, which started to limit the charge generation efficiency. The obvious inconsistency in the required energy between for PHJs in this study and for the some efficient BHJs (less than 0.1 eV) could be due to the difference in the interfacial structures; there could be key features in the interfacial structures of BHJs that universally reduce the apparent thresholds for the charge separation. Revealing these factors will lead to find the methodology for reducing the energy loss while maintaining efficient charge generation.

## Methods

**Materials**. Regioregular poly(3-hexylthiophene-2,5-diyl) (P3HT, RMI-001EE, $M_w$: 36,000, Rieke Metals), poly({4,8-bis[(2-ethylhexyl)oxy]benzo[1,2-b:4,5-b']dithiophene-2,6-diyl}{3-fluoro-2-[(2-ethylhexyl)carbonyl]thieno[3,4-b]thiophenediyl}) (PTB7, 1-Material), and [6,6]-phenyl C$_{61}$ butyric acid methyl ester (PC$_{61}$BM, purity: 99.5%, Solenne) were used as received. Poly[5,5'-bis(2-butyloctyl)-(2,2'-

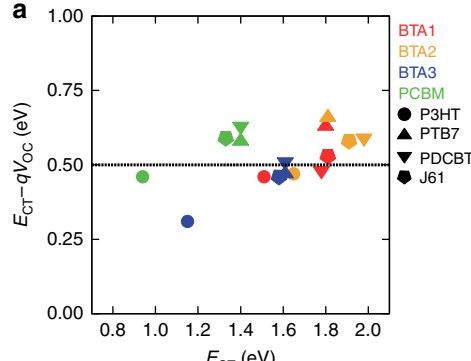
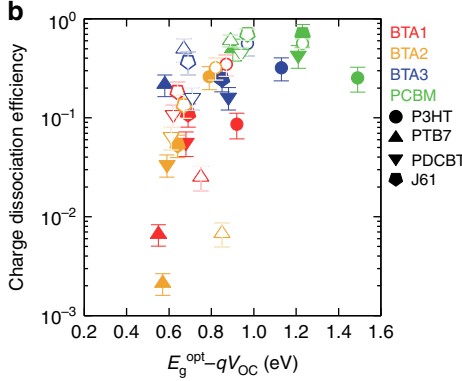

**Fig. 7** Energy loss. **a** $E_{CT} - qV_{OC}$, the energy loss due to charge recombination plotted against $E_{CT}$. **b** Charge generation efficiency plotted against the overall energy loss, $E_g^{opt} - qV_{OC}$. The error bar indicates the variation of the assumed exciton collection length

bithiophene)-4,4′-dicarboxylate-alt-5,5′-2,2′-bithiophene] (PDCBT, $M_w$: 128,000), J61, and the BTA series were synthesized following literature methods[28–30].

**Device fabrication**. A glass substrate with a patterned ITO electrode was cleaned by sequential ultrasonication in detergent solution, water, 2-propanol, and acetone, followed by UV-O$_3$ treatment. The ITO surface was spin-coated with a PEIE buffer layer approximately 10 nm thick. The acceptor material was spin-coated onto the PEIE layer. An aqueous solution of poly(sodium 4-styrenesulfonate) (PSS; 30 mg mL$^{-1}$; $M_w$: 70,000; Sigma-Aldrich) was spin-coated onto a pre-cleaned glass substrate at 3000 rpm for 30 s. A donor polymer was spin-coated onto the glass/PSS substrates. Details of the spin-coating conditions and film thicknesses are summarized in Supplementary Table 2. The glass/PSS/polymer substrate was gently placed upside down on the substrate with the ITO/PEIE/acceptor, and one drop of water was placed on the edge of these two substrates. Water selectively penetrated and dissolved the PSS layer, allowing the polymer layer to be transferred onto the bottom acceptor film. Photographs of the film transfer process have been published previously[34]. A MoO$_3$ hole-transporting layer (7.5 nm) and Ag electrodes (100 nm) were deposited by thermal evaporation through a metal mask under high vacuum (~10$^{-4}$ Pa). All samples were encapsulated with a glass cap and UV-curable resin in a dry N$_2$-filled glovebox

**UPS and LEIPS measurements**. ITO substrates were used for the UPS and LEIPS measurements. UPS was performed with a photoelectron spectroscopy system (PHI5000 VersaProbe II, ULVAC-PHI Inc.) with He I excitation (21.2 eV). For all UPS measurements, a −5.0 V bias was applied to the samples. Detail of the LEIPS setup has been described elsewhere[61]. The samples were irradiated with an electron beam in a vacuum chamber with pressure below 1 × 10$^{-7}$ Pa. The kinetic energies of the incident electrons were 0–4 eV to avoid sample damage, and the electron current densities were between 10$^{-6}$ and 10$^{-5}$ A cm$^{-2}$. The emitted photons were collected by a photon detector equipped with an optical band-pass filter and a photomultiplier tube. The LEIPS spectrum was obtained by scanning the electron kinetic energy. To minimize the uncertainty in the electron affinity, the spectra were taken at different center wavelengths of the band-pass filter of 260, 285, and 335 nm.

**Current-voltage characteristics**. The $J$-$V$ characteristics of the devices were measured under simulated solar illumination (AM 1.5, 100 mW cm$^{-2}$) from a solar simulator with a 150 W Xe lamp (PEC-L11, Peccell Technologies). The light intensity was calibrated with a standard silicon solar cell (BS520, Bunkoh-Keiki). The active area of each device was defined by using a 0.12 cm$^2$ metal mask.

**EQE measurements**. The EQE of each device was measured with monochromatic light (SM-250F, Bunkoh-Keiki). The light intensity was calibrated with a standard Si and InGaAs photodetector. The photocurrent was recorded using a lock-in amplifier (LI5640, NF) with a low-noise current amplifier (DLPCA-200, FEMTO). The lock-in frequency was 85 Hz. For measurements with a reverse bias, the DC voltage output of the current amplifier was used.

**X-ray reflectivity measurements**. X-ray reflectivity was performed with an X-ray diffractometer (Smartlab, Rigaku), and the reflection patterns were fitted by using GlobalFit software (Rigaku). Monochromatized Cu Kα radiation (λ = 0.154 nm) was generated at 45 kV and 200 mA. Films were prepared on a Si/SiO$_2$ substrate using the same spin-coating conditions as for the photovoltaic devices.

**Light absorption measurements**. The spectral absorptance of the donor and the acceptor films in transmittance mode were measured with a UV-Vis spectrophotometer (V-670, JASCO) from 400 to 900 nm. Reflectance mode with an integrating sphere was used to measure the reflectance and transmittance of the devices, and the device absorptance was calculated by subtracting the measured reflectance and transmittance from 1.

**Light intensity and temperature dependence of $J$-$V$ characteristics**. Devices were placed in a stainless-steel chamber filled with dry N$_2$ (Kitano Seiki). The light source was a 5 W warm white light-emitting diode (LED) (XP-G2, Cree) with a homemade condensing lens system. The LED output power was controlled so that irradiated devices exhibited nearly identical performance to that under AM1.5, 100 mW cm$^{-2}$ solar light. Light intensity was altered by a combination of two neutral-density filters. The intensity was calibrated using a standard silicon photodiode. Temperatures at both the sample stage and the device surface were measured by thermocouples.

**PL and EL measurements**. PL and EL were measured by using a spectrofluorometer equipped with a photomultiplier for the visible region and a liquid N$_2$-cooled InGaAs detector for the infrared region (Nanolog, Horiba). A bias voltage was applied to the devices and the current was recorded using a source measurement unit (2400, Keithley).

**Ellipsometry measurements and transfer matrix optical simulations**. Ellipsometry measurements were conducted on an ellipsometer (RC2-UI, J.A. Woollam) in the wavelength range of 210 to 1690 nm with incident angles of 45° to 75°. The thin films of the organic materials were prepared on a fused silica substrate by spin-coating. The transmittance of each film was measured at the same time and was used to determine the optical constants. The isotropic optical model was used for the PCBM thin film, whereas out-of-plane anisotropic models were used to analyze the data for the other organic materials. A gradient component model in the vertical direction was used for the ITO. Optical simulations of the bilayer OSC devices were performed based on the previously reported transfer matrix model[38,39]. The film thickness of each layer was measured by X-ray reflectivity. The electric field distributions and energy dissipations of monochromatic light were calculated for all the locations in each layer. The unit of the layer thickness and the wavelength were set to 1 nm. The simulations were performed by using lab-made code run on MATLAB.

**TPC measurements**. An LED bias light with neutral-density filters was used. Perturbation was performed with an N$_2$-dye pulse laser (KEC-160, Usho) with an excitation wavelength, repetition rate, and pulse duration of 532 nm, 100 Hz, and 0.4 ns, respectively. Resistance (50 Ω) was put parallel to the input of a digital oscilloscope (DS-5632, Iwatsu), and the transient current was calculated using Ohm's law. The integral of the transient current over time provided the amount of transient charge.

## Data availability
The authors declare that the main data supporting the findings of this study are available within the article and its Supplementary Information files. Extra data are available from the corresponding author upon reasonable request.

## Code availability
The MATLAB code and the optical constants for the optical simulations are provided as a Source Data file.

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

## Acknowledgements

We thank Mr. Takanobu Sato (J. A. Woollam Japan) for conducting the ellipsometry measurements. This research was supported in part by JSPS KAKENHI (JP18K14301), JST ALCA (JPMJAL1404) and the Futaba Research Grant Program of the Futaba Foundation.

## Author contributions

K.N. designed the experiments and measured the J-V characteristics, EQE, light intensity dependence, temperature dependence, TPC, UPS, light absorption, EL, and PL. Y.C. prepared all PHJ devices and pristine films and measured the X-ray reflectivity. H.Y. developed the LEIPS system and supervised the LEIPS experiments. W.H. performed the LEIPS measurements. J.H. synthesized PDCBT. E.Z. and B.X. synthesized BTAs. K.T. supervised the overall project. K.N. and K.T. prepared the manuscript. All the authors reviewed the final manuscript.

## Additional information

**Competing interests:** The authors declare no competing interests.

