## [Peer Review File · Nature Communications]

Reviewers' comments:

Reviewer #1 (Remarks to the Author):

Nakano and coworkers provide a comprehensive and systematic study of energy losses in organic hetero junctions. The novelty lies in the preparation of the hetero junctions. In contrast to the majority of studies that focus on bulk hetero junctions, Nakano and coworkers prepare and study what they call 'planar heterojunction'. In the latter type of junctions, extensive mixing of the donor and acceptor materials that are used in the hetero junction is precluded. Arguably this leads to a simplification of the problem of charge generation and certainly gives a new experimental perspective.

The study is state of the art involving the most modern materials and techniques and provides new insight valuable to a large community of scientists interested in photo voltaic energy conversions. Yet with regards to the following point the manuscript is not yet sufficiently clear for me to recommend immediate publication

1) The authors determine the energy of the Charge Transfer state (E_{CT}) by extrapolating the dependence of V_{OC} as function of temperature to the absolute zero. This method essentially relies on the assumption that at zero temperature non-radiative decay no longer occurs. The latter assumption is however questionable for molecular systems. Many dye molecules with vanishing fluorescence quantum yield due to rapid non-radiative deactivation also remain non-fluorescent at the lowest reachable temperatures. There it would be good to list the E_{CT} obtained in the supporting information and compare the values for E_{CT} with those determined for measurement of the charge transfer absorption and charge transfer luminescence.

2) The authors argue that losses due to non-geminate recombination in the planar hetero junctions studied are negligible. The authors indeed provide credible experimental data to support their claim. Yet the data presented pertain to short circuit conditions. What matters most is the behavior for applied bias voltages close to the open circuit. In the cells studied the fill factor is well below unity and so for bias voltages close to the open circuit conditions, substantial losses occur. These losses may be due to reduced charge generation efficiency or to non-geminate recombination. The authors need to justify why they think that also under load (i.e. for bias voltages close to V_{OC}), the non-geminate recombination losses are still negligible.

3) The authors comment on the difference between the threshold criterion they derive for the planar junctions ($E_{g, opt} - E_{CT} > 0.2-0.3\text{eV}$) and the criterion as it has been obtained for bulk hetero junctions ($E_{g, opt} - E_{CT} < 0.1\text{eV}$) They ascribe the difference in the value of the threshold energy to the variation in the disorder. Although this makes sense, the analysis and discussion (see page 19, line 256) are not completely satisfactory because if both $E_{g, opt}$ and E_{CT} become larger due to disorder, the difference between these numbers would not necessarily decrease for the more disordered type of junction

The authors should try to pinpoint more accurately what they think. I guess that in order to explain the data, E_{CT} should rise more steeply with increasing disorder than $E_{g, opt}$. Could one, based on this argument also expect higher V_{OC} values in the bulk hetero junction in comparison with the planar junctions. The data presented seem to hint at this. Please comment.

4) In their conclusion, the authors argue that when their criterion $E_{g, opt} -$

$E_{CT} > 0.2-0.3$ eV is the major loss process that becomes activated is decay of the lowest excited singlet state (S1) back to the ground state. Several studies have pointed out that the S1 state might also decay to a lower lying triplet state instead of going to the ground state. The authors should assess whether this additional loss channel may also be relevant for the hetero junctions under study here.

5) The main conclusion of the authors is the existence of an energy criterion ($E_{g}^{opt} - E_{CT} > 0.2-0.3$ eV) for efficient photovoltaic energy conversion in organic hetero junctions. Yet the implications of this newly established criterion for the limits of power conversion efficiency by organic solar cells remain unclear. Now the authors seem to indicate that bulk hetero junctions have intrinsically a higher internal efficiency because of the lower $E_{g}^{opt} - E_{CT}$. Yet as the authors argue, this 'better' performance results from more disorder. Then if the disorder is high the absorption onset for the bulk hetero junctions should not be as steep as for the planar junctions which would imply that the bulk hetero junctions have higher optical losses and lower external efficiency from photons not getting absorbed for photon energies close or at the optical gap. The authors should briefly outline in more detail the implications of their criterion for any upper limit to power conversion efficiency.

Stefan Meskers

Reviewer #2 (Remarks to the Author):

In the manuscript from K. Tajima and colleagues entitled "Anatomy of the Energetic Driving Force for Charge Generation in Organic Solar Cells" the authors have systematically investigated 16 donor/acceptor (D/A) combinations in planar heterojunction organic solar cells (OSCs). The manuscript analyses the energy and internal quantum efficiency losses associated with photovoltaic energy conversion, particularly with respect to the energetic driving force at the D/A interface governing charge separation.

For a number of years, the prevailing notion in the OSC community was based on the widely known empirical rule that in most D/A combinations efficient charge separation occurs only when the energetic driving force, i.e. the energetic difference between the lowest energy singlet state (E_{S1}) and the energy of the charge-transfer state (E_{CT}), is ~ 0.3 eV or larger, otherwise charge separation becomes inefficient and generally increasingly field-dependent. The physics of this are not fully understood despite significant efforts. In the manuscript by K. Tajima et al. the authors employ a similar/related methodology to that published earlier by other authors (e.g. DOI: 10.1002/adma.201305283 and DOI: 10.1021/ja5131897 etc.) and their results show that the threshold energy difference between E_{S1} and E_{CT} for efficient charge separation is $\sim 0.2-0.3$ eV also in their investigated materials.

While the authors study is systematic and sound for the most part, my main criticism is that besides providing sensible (albeit already known, e.g. DOI: 10.1021/acsami.8b12077 and there are many more) qualitative explanations for the "0.3 eV rule" the author's work does not explain the underlying mechanism governing this number. Due to this reason alone, the manuscript lacks sufficient novelty and conceptual advance to be considered for publication in such a high impact journal as Nature Communications. In addition, there are known exceptions to the "0.3 eV rule", e.g. DOI: 10.1002/adfm.201200608, DOI: 10.1002/adma.201405485 and DOI: 10.1002/adma.201504417, which we, as a community, should strive for in state-of-the-art materials to minimize open-circuit voltage (V_{oc}) losses for improved OSC efficiency. However, the submitted manuscript does not offer

conceptual insight regarding the origin of such small losses, thus not only significantly limiting its immediate interest to the OSC community, but also to researchers from related disciplines.

Having said that, the author's study appears systematic and would be a useful addition to the literature, although I would suggest a more specialized journal. Below I provide additional comments that I think the authors should address:

- In the abstract the authors state: "...we did not observe any correlation of the charge generation efficiency with the energy difference between the CT and the charge-separated states, indicating that the binding of the charge pairs in the CT state is less important for the charge generation." I do not think the authors can conclude this from their data. Measuring the energies of the charge separated states (E_{CS}) is non-trivial due to the disordered nature of organic semiconductors (which is ignored in this manuscript), resulting in a wide spread of charge-separated state energies and kinetic effects, e.g. see DOI: 10.1002/adfm.201200249 and DOI: 10.1002/aenm.201301743. This could be one, among the many reasons for the lack of distinct correlation between IQE and $E_{CS} - E_{CT}$. I think it would more appropriate to state that "... the effect of the CT pair binding energy is not apparent and convoluted by other effects" and discuss the possible influence of energetic disorder on the observed trends. How are the results affected by energetic disorder?

- The authors state that: "... planar heterojunctions (PHJs) are a suitable structure for investigating the direct correlation between the interfacial properties and the device performances, because of their well-defined interfacial structure". It would be helpful if the authors comment on the evidence that the interfacial structure is in fact well defined.

- To estimate E_{CT} , the authors rely on measuring the V_{oc} dependence versus temperature (T), however, the corresponding data are not shown in the manuscript. I think the authors must show the data, e.g. in the Supporting information, since the V_{oc} vs T dependence can be non-linear, e.g. see DOI: 10.1103/PhysRevLett.114.128701, rendering the fits ill-defined.

- Line 145: "IQE was evaluated by dividing JSC by the total flux of absorbed photons, which were calculated by using the AM1.5 solar spectrum and the spectral absorptance of the devices measured with an integrating sphere." This procedure is prone to error due to thin-film interference effects in the OSC device stack. I think the authors should provide a compelling argument why optical modelling was not employed for accurate IQE determination.

- The authors should show their device current-voltage (IV) curves in the Supporting Information. In addition, showing external and internal quantum efficiency (EQE and IQE, respectively) spectra would be useful.

- Some of the photoluminescence (PL) data in Figure 5 and Figure S6 appear to have a poor background correction, c.f. Figure 5b spectra at energies >1.8 eV.

- The authors mostly focus on the energetic driving force that is the smallest in the D/A blend, i.e. the focus is on the smallest ($E_{S1} - E_{CT}$), e.g. for electron transfer. However, hole transfer will still occur, albeit at a different (in this case larger) $E_{S1} - E_{CT}$. I think the authors should elaborate on how their results are affected by the fact that both electron and hole transfer occur at the D/A interface in their measurements, albeit at different driving forces, i.e. each blend has two driving forces, not one. For example, when discussing the PL experiments in Figure 5, only one driving force is considered, whereas at the excitation wavelength of 540 nm both D/A materials are photoexcited, c.f. Figure S3.

Reviewer #3 (Remarks to the Author):

The authors study the internal quantum yield and field-dependence of charge generation in 16 combinations of polymers with fullerene and non-fullerene acceptors in planar organic solar cells. They relate this to the driving force (difference of S1 energy to CT state energy or CS state energy) and find that high field-independent IQEs require a S1-CT difference of more than 0.2-0.3 eV, while the charge separation driving force (CT-CS) is not relevant. I-V characterization at different temperatures and light intensities, absorption/emission spectroscopy and UPS/LEIPS are carried out to determine the experimental parameters. The study is indeed very interesting to the OPV and electron transfer communities and could in principle be published in Nature Communications. There are however a number of conclusions that are currently not well enough substantiated, so that the manuscript needs major revisions before being re-considered for publication:

- The authors determine the energy levels by photoelectron, absorption and emission spectroscopy on the neat films and assume that they are the same in the PHJ devices. It is true that no intermixing or complicated morphologies occur in the PHJ structure, but the energy levels could still be affected by donor-acceptor interactions and electric fields at the interface. Could the authors comment on this?
- All driving forces reported in the manuscript are related to enthalpy, while it is actually the free energy (enthalpy and entropy) that determines if the processes are spontaneous. Do the authors think that entropy could play a role in their PHJ samples?
- The CT state energy is obtained here by the temperature-dependent IV characteristics. Very often, sensitive EQE measurements (and EL) are used instead. Could the authors comment why they chose their method and if it is better?
- The authors always use the S1 energy of the low bandgap component to define the S1-CT driving force (except PCBM). But then the IQE is spectrally integrated (i.e. taken from the JSC, solar spectrum and absorptance) and therefore refers to the photocurrent generated by both the electron and hole transfer processes, which have two different driving forces. It would be much more correct to obtain the IQE in a wavelength-resolved way and to relate the smaller driving force only to IQE at wavelengths where the low band gap component selectively absorbs.
- Measuring IQE is a tricky task, usually requiring complex transfer matrix modelling of the photon densities inside the device. Can the authors comment on whether obtaining the absorptance in an integrating sphere is enough to get reliable results?
- For the electric field dependence, it is not clear how the "ratio of charge density" was obtained. Is it the ratio of IQEs?
- For the bias-dependent emission, it would be important to know more precisely which of the donor or acceptor is emitting, and to control which one was excited. Now this is left open. If the emission is always from the low bandgap component (also if the high bandgap one is excited), the S1 state is formed by CT back-recombination. In this case, the field impacts the CT state. If the emission is always from the excited component, it means that the field impacts the charge transfer from the S1 state.

Point-by-point responses to reviewers' comments (the original comments are in *italic*):

Reviewers' comments:

Reviewer #1 (Remarks to the Author):

Nakano and coworkers provide a comprehensive and systematic study of energy losses in organic hetero junctions. The novelty lies in the preparation of the hetero junctions. In contrast to the majority of studies that focus on bulk hetero junctions, Nakano and coworkers prepare and study what they call 'planar heterojunction'. In the latter type of junctions, extensive mixing of the donor and acceptor materials that are used in the hetero junction is precluded. Arguably this leads to a simplification of the problem of charge generation and certainly gives a new experimental perspective.

The study is state of the art involving the most modern materials and techniques and provides new insight valuable to a large community of scientists interested in photo voltaic energy conversions.

Yet with regards to the following point the manuscript is not yet sufficiently clear for me to recommend immediate publication.

1) The authors determine the energy of the Charge Transfer state (ECT) by extrapolating the dependence of VOC as function of temperature to the absolute zero. This method essentially relies on the assumption that at zero temperature non-radiative decay no longer occurs. The latter assumption is however questionable for molecular systems. Many dye molecules with vanishing fluorescence quantum yield due to rapid non-radiative deactivation also remain non-fluorescent at the lowest reachable temperatures. There it would be good to list the ECT obtained in the supporting information and compare the values for CT with those determined for measurement of the charge transfer absorption and charge transfer luminescence.

First, we thank the reviewer for the insightful comments.

We conducted highly sensitive EQE and EL measurements to evaluate the absorption and emission of the CT states in all the systems. For four PHJ systems out of the 16, we observed well-resolved CT bands and obtained E_{CT} values using the optical measurements with reasonable fittings. However, six systems had lower goodness of fit due to the deformation of the peak shapes, and the other six systems had poorly resolved E_{CT} bands owing to the S_1 bands and it was impossible to extract information about the CT states. The extracted values are summarized in Table S3.

The E_{CT} values determined from the $qV_{OC}-T$ plots and EQE/EL measurements matched well for eight systems with deviations below 15% (Fig. S9 and Table S3), whereas two PHJ systems with low fitting reliability showed larger deviations (47% and 30% for BTA3/P3HT and PCBM/P3HT, respectively). We think that the accuracy of the EQE/EL technique may be limited in PHJs because the small D/A interface area leads to a smaller signal from interfacial absorption/emission relative to those from the bulk, which reduced the reliability of the gaussian fittings. Because of the lower reliability of the E_{CT} values determined by the EQE/EL measurements, we decided to use the E_{CT} values from the $qV_{OC}-T$ plots primarily for discussion. We confirmed that even when we used the E_{CT} values obtained by EQE/EL measurements, our discussion and conclusion remained unchanged (Fig. S12).

This discussion has been included in the revised manuscript (“Evaluation of E_{CT} ” section).

2) The authors argue that losses due to non-geminate recombination in the planar hetero junctions studied are negligible. The authors indeed provide credible experimental data to support their claim. Yet the data presented pertain to short circuit conditions. What matters most is the behavior for applied bias voltages close to the open circuit. In the cells studied the fill factor is well below unity and so for bias voltages close to the open circuit conditions, substantial losses occur. These losses may be due to reduced charge generation efficiency or to non-geminate recombination. The authors need to justify why they think that also under load (i.e. for bias voltages close to V_{OC} , the non-geminate recombination losses are still negligible.

We did not intend to suggest that non-geminate recombination losses are negligible under load. The light intensity dependence of J_{SC} and the charge collection measurements can only tell us that the effect of non-geminate recombination is negligible under short-circuit conditions and reverse bias conditions. Therefore, we mainly discussed the charge generation efficiency and electric field dependence of charge generation from the short-circuit to reverse bias conditions (J ratios under biases of 0 and -1 V).

We agree with the reviewer that non-geminate recombination also affects the FF. The FF of the PHJs also decreased with $E_g^{opt} - E_{CT}$ below about 0.3 eV (Fig. S14), whereas a similar but weaker trend was observed in the relationship with $E_{CS} - E_{CT}$. The trend for $E_g^{opt} - E_{CT}$ could also suggest that $E_g^{opt} - E_{CT}$ has a large effect on the electric field dependence of the charge generation. However, both the geminate and non-geminate recombination can affect the FF and their complete separation would be difficult under the OSC operating conditions.

The corresponding text in “Electric field dependence of charge separation” has been revised according to the discussion above.

3) The authors comment on the difference between the threshold criterion they derive for the planar junctions ($E_{g}^{opt} - E_{CT} > 0.2-0.3\text{eV}$) and the criterion as it has been obtained for bulk hetero junctions ($E_{g}^{opt} - E_{CT} < 0.1\text{eV}$). They ascribe the difference in the value of the threshold energy to the variation in the disorder. Although this makes sense, the analysis and discussion (see page 19, line 256) are not completely satisfactory because if both E_{g}^{opt} and E_{CT} become larger due to disorder, the difference between these numbers would not necessarily decrease for the more disordered type of junction.

The authors should try to pinpoint more accurately what they think. I guess that in order to explain the data, E_{CT} should rise more steeply with increasing disorder than E_{g}^{opt} . Could one, based on this argument, also expect higher VOC values in the bulk hetero junction in comparison with the planar junctions. The data presented seem to hint at this. Please comment.

We have illustrated the possible effects of interfacial disorder in Fig. S17. If only the donor phase had a disordered layer, E_{g}^{opt} of the layer would be increased and E_{CT} would be increased because of the downward shift of E_{HOMO}^D . For the charge generation process at the interfaces of the disordered donor and the acceptor, the energetic driving force (i.e., $E_{g}^{opt} - E_{CT}$) must be calculated by using E_{g}^{opt} of the disordered layer. However, this broadened, blue-shifted light absorption of the thin disordered layer may overlap with the light absorption of the bulk of the donor. Moreover, the absorption of the disordered layer is expected to be much smaller than that of the bulk because it is a thin layer at the D/A interface. Therefore, the light absorption

of the disordered layer at the interface was not detected in the absorption spectra. Consequently, although the E_g^{opt} values for the pristine films or the bulk of the donor are usually used for estimating $E_g^{\text{opt}} - E_{\text{CT}}$, they could lead to the underestimation of $E_g^{\text{opt}} - E_{\text{CT}}$ in BHJs.

The explanation has been clarified in “Discussion: Efficient OSCs with small energy offset” and the caption of Fig. S17.

4) In their conclusion, the authors argue that when their criterion $E_{\text{gopt}} - E_{\text{CT}} > 0.2-0.3 \text{ eV}$ is the major loss process that becomes activated is decay of the lowest excited singlet state (S_1) back to the ground state. Several studies have pointed out that the S_1 state might also decay to a lower lying triplet state instead of going to the ground state. The authors should assess whether this additional loss channel may also be relevant for the hetero junctions under study here.

The intersystem crossing from S_1 to T_1 is an additional loss process that we did not consider in this manuscript. However, our main conclusions about the charge generation efficiency would not be affected by this process; the additional decay path would appear only as a change in the S_1 lifetime. The efficiency of the process could be affected if there is an efficient CT process from the decayed T_1 to CT^3 state. However, this process is unlikely for the systems of interest with small $E_g^{\text{opt}} - E_{\text{CT}}$ of $<0.3 \text{ eV}$ (close to the threshold) considering the typical electron correlation energy between S_1 and T_1 .

5) The main conclusion of the authors is the existence of an energy criterion ($E_{\text{gopt}} - E_{\text{CT}} > 0.2-0.3 \text{ eV}$) for efficient photovoltaic energy conversion in organic hetero junctions. Yet the implications of this newly established criterion for the limits of power conversion efficiency by organic solar cells remain unclear. Now the authors seem to indicate that bulk hetero junctions have intrinsically a higher internal efficiency because of the lower $E_{\text{gopt}} - E_{\text{CT}}$. Yet as the authors argue, this 'better' performance results from more disorder. Then if the disorder is high the absorption onset for the bulk hetero junctions should not be as steep as for the planar junctions which would imply that the bulk hetero junctions have higher optical losses and lower external efficiency from photons not getting absorbed for photon energies close or at the optical gap. The authors should briefly outline in more detail the implications of their criterion for any upper limit to power conversion efficiency.

The minimum driving force for the efficient charge generation in OSCs strongly affects the limit of the power conversion efficiency. The current molecular design depends on this value

and many groups are trying to match the energy level of their materials as closely as possible with the optimum value of <0.1 eV. However, we suggest in this work that the efficient charge transfer with very small $E_g^{\text{opt}} - E_{\text{CT}}$ observed for the mixed BHJs could be due to the interfacial disordered layers; the discrepancy in the criteria for PHJs and BHJs highlights the importance of controlling interfacial structures. This concept could change the design principles of organic semiconducting materials beyond simple energy level matching.

This discussion was added to the “Discussion: Efficient OSCs with small energy offset” section.

Reviewer #2 (Remarks to the Author):

In the manuscript from K. Tajima and colleagues entitled "Anatomy of the Energetic Driving Force for Charge Generation in Organic Solar Cells" the authors have systematically investigated 16 donor/acceptor (D/A) combinations in planar heterojunction organic solar cells (OSCs). The manuscript analyses the energy and internal quantum efficiency losses associated with photovoltaic energy conversion, particularly with respect to the energetic driving force at the D/A interface governing charge separation.

For a number of years, the prevailing notion in the OSC community was based on the widely known empirical rule that in most D/A combinations efficient charge separation occurs only when the energetic driving force, i.e. the energetic difference between the lowest energy singlet state (E_{S1}) and the energy of the charge-transfer state (E_{CT}), is ~ 0.3 eV or larger; otherwise charge separation becomes inefficient and generally increasingly field-dependent. The physics of this are not fully understood despite significant efforts. In the manuscript by K. Tajima et al. the authors employ a similar/related methodology to that published earlier by other authors (e.g. DOI: 10.1002/adma.201305283 and DOI: 10.1021/ja5131897 etc.) and their results show that the threshold energy difference between E_{S1} and E_{CT} for efficient charge separation is ~ 0.2 - 0.3 eV also in their investigated materials.

While the authors study is systematic and sound for the most part, my main criticism is that besides providing sensible (albeit already known, e.g. DOI: 10.1021/acsami.8b12077 and there are many more) qualitative explanations for the "0.3 eV rule" the author's work does not explain the underlying mechanism governing this number. Due to this reason alone, the manuscript lacks sufficient novelty and conceptual advance to be considered for publication in such a high impact journal as Nature Communications. In addition, there are known exceptions to the "0.3 eV rule", e.g. DOI: 10.1002/adfm.201200608, DOI: 10.1002/adma.201405485 and DOI: 10.1002/adma.201504417, which we, as a community, should strive for in state-of-the-art materials to minimize open-circuit voltage (V_{oc}) losses for improved OSC efficiency. However, the submitted manuscript does not offer conceptual insight regarding the origin of such small losses, thus not only significantly limiting its immediate interest to the OSC community, but also to researchers from related disciplines.

First, we thank the reviewer for the insightful comments.

In the previous version of the manuscript, we decided not to include a detailed discussion of the mechanism of the observed energy dependence because the measured IQEs plotted against

$E_g^{\text{opt}} - E_{\text{CT}}$ showed saturated behavior in the large $E_g^{\text{opt}} - E_{\text{CT}}$ region, where Marcus theory predicts that the charge transfer rate decreases due to the inverted region. In the revised manuscript, we performed optical modeling of the light absorption and evaluated the charge generation efficiency separately from the excitons in the donor and the acceptor sides. We found that the charge generation efficiency started to decrease at large $E_g^{\text{opt}} - E_{\text{CT}}$ values (>0.8 eV), which was consistent with Marcus theory. We have included the above discussion in the revised manuscript.

As for the impact on the OSC research community, the minimum driving force for the efficient charge generation in OSCs strongly affects the limit of power conversion efficiency. The current molecular design depends on this value and many groups are trying to match the energy level of their materials as closely as possible with the most optimistic value of <0.1 eV. However, based on the observed results for PHJs in this work we suggest in this work that the efficient charge transfer with very small $E_g^{\text{opt}} - E_{\text{CT}}$ observed for the mixed BHJs could be due to the interfacial disordered layers; the discrepancy in the criteria for PHJs and BHJs highlights the importance of controlling interfacial structures. This concept could change the design principles of organic semiconducting materials beyond simple energy level matching. This discussion was added to the “Discussion: Efficient OSCs with small energy offset” section.

In sum, the main novelty of this work resides on the demonstration of the general “0.3 eV rule” with the ambiguously defined state energies in PHJs without variations of the interfacial structures, and the conceptual advance based on this observation is that the efficient charge generation previously observed with a small energy loss for BHJs could be ascribed to the structural factors near the D/A interface. Thus, we believe the manuscript will be of interest to both the OSC community and the broader readership of *Nature Communications*.

Having said that, the author's study appears systematic and would be a useful addition to the literature, although I would suggest a more specialized journal. Below I provide additional comments that I think the authors should address:

- In the abstract the authors state: “...we did not observe any correlation of the charge generation efficiency with the energy difference between the CT and the charge-separated states, indicating that the binding of the charge pairs in the CT state is less important for the charge generation.” I do not think the authors can conclude this from their data. Measuring the energies of the charge separated states (E_{CS}) is non-trivial due to the disordered nature

of organic semiconductors (which is ignored in this manuscript), resulting in a wide spread of charge-separated state energies and kinetic effects, e.g. see DOI: 10.1002/adfm.201200249 and DOI: 10.1002/aenm.201301743. This could be one, among the many reasons for the lack of distinct correlation between IQE and $E_{CS} - E_{CT}$. I think it would more appropriate to state that "... the effect of the CT pair binding energy is not apparent and convoluted by other effects" and discuss the possible influence of energetic disorder on the observed trends. How are the results affected by energetic disorder?

We agree that our previous description of the relationship between IQE and $E_{CS} - E_{CT}$ may be too strong. We modified the statement in the abstract as the reviewer suggested. The effects of the structural disorder at the interfaces and in the bulk are important and will be investigated in the future, but it is beyond the scope of this manuscript.

- The authors state that: "... planar heterojunctions (PHJs) are a suitable structure for investigating the direct correlation between the interfacial properties and the device performances, because of their well-defined interfacial structure". It would be helpful if the authors comment on the evidence that the interfacial structure is in fact well defined.

In previous publications, we have reported the preparation of PHJ films based on several semiconducting polymers/PCBM bilayers by the film transfer method (DOI: 10.1038/srep29529 and 10.1038/nmat3026). X-ray reflectivity (XRR) of the bilayer films showed that the interference fringes were well reproduced based on the thicknesses and densities of the donor and acceptor films that were measured before the film transfer process, indicating the clean formation of the D/A interface. This is because the film transfer method requires mild conditions (room temperature, no pressure or heat treatment, no organic solvents), preventing the intermixing of the layers. Therefore, we believe that the interfacial structures were preserved well. We also conducted XRR measurements on the PHJs of polymer/non-fullerene acceptors in this study. However, unlike the polymer/PCBM PHJs, the material densities of the donors and the acceptors were too close to give sufficient contrast in the X-ray reflection.

- To estimate E_{CT} , the authors rely on measuring the V_{oc} dependence versus temperature (T), however, the corresponding data are not shown in the manuscript. I think the authors must show the data, e.g. in the Supporting information, since the V_{oc} vs T dependence can be non-linear, e.g. see DOI: 10.1103/PhysRevLett.114.128701, rendering the fits ill-defined.

We added V_{OC} - T plots for all the devices to Fig. S7. The qV_{OC} - T plots for all the combinations of materials showed a linear relationship in the range of 300–210 K (Fig. S7), as in previous reports.

- Line 145: “IQE was evaluated by dividing JSC by the total flux of absorbed photons, which were calculated by using the AM1.5 solar spectrum and the spectral absorptance of the devices measured with an integrating sphere.” This procedure is prone to error due to thin-film interference effects in the OSC device stack. I think the authors should provide a compelling argument why optical modelling was not employed for accurate IQE determination.

According to the reviewer’s suggestion, we performed transfer matrix optical simulations for all the PHJ devices to calculate the absorptance of the organic layers without the parasitic absorption of Ag and ITO/glass substrate. By using this information, we reproduced the external quantum efficiency spectra of the devices. Based on an exciton diffusion model with an appropriate exciton collection length near the D/A interface, the charge generation efficiencies at the interface from the S_1 to CT states were evaluated separately for the donor and the acceptor. This analysis doubled the number of data points from 16 to 32 in the key figure (Fig. 4 in the revised manuscript). The charge generation efficiency decreased drastically as $E_g^{opt} - E_{CT}$ decreased. This is the same conclusion as in the previous version of the manuscript, but the discussion based on the charge generation efficiency is more physically meaningful than that based on the IQEs. There was a decrease in the efficiency at larger $E_g^{opt} - E_{CT}$ of >0.8 eV, which is explained by Marcus theory.

The discussion above was added to the “Evaluation of interfacial charge generation efficiency” and “Relationship between the state energy difference and charge generation efficiency” sections.

- The authors should show their device current-voltage (IV) curves in the Supporting Information. In addition, showing external and internal quantum efficiency (EQE and IQE, respectively) spectra would be useful.

We added J - V curves under AM1.5 100 mW/cm² simulated solar light irradiation and the external quantum efficiency for all devices in Figs. S5 and S6, respectively. In the revised version, we used the charge generation efficiency at the D/A interfaces determined by reproducing EQE based on the exciton diffusion model. We have added the measured and simulated EQE spectra in Fig. S10.

- Some of the photoluminescence (PL) data in Figure 5 and Figure S6 appear to have a poor background correction, c.f. Figure 5b spectra at energies >1.8 eV.

We found that there was a weak emission from the black tape used to attach the sample to the sample holder, which produced the background artifact. We modified the experimental setup and performed the photoluminescence measurements again, which solved the background problem. We have updated PL data in the revised manuscript (Fig. 6).

- The authors mostly focus on the energetic driving force that is the smallest in the D/A blend, i.e. the focus is on the smallest ($E_{S1} - E_{CT}$), e.g. for electron transfer. However, hole transfer will still occur, albeit at a different (in this case larger) $E_{S1} - E_{CT}$. I think the authors should elaborate on how their results are affected by the fact that both electron and hole transfer occur at the D/A interface in their measurements, albeit at different driving forces, i.e. each blend has two driving forces, not one. For example, when discussing the PL experiments in Figure 5, only one driving force is considered, whereas at the excitation wavelength of 540 nm both D/A materials are photoexcited, c.f. Figure S3.

As the reviewer pointed out, the charge separation efficiencies can depend on whether the events happen at the donor or acceptor side. To address this, as in the response to the above comment, the charge generation efficiency at the interface from the S_1 to CT states was evaluated separately for the donor and acceptor in the revised manuscript. The same trend for the donor and acceptor excitation was observed, indicating that the dependence of the energetic driving force is similar for the S_1 states of the donor and the acceptor.

In the above analysis, we used different E_g^{opt} values for the donors and the acceptors to calculate $E_g^{\text{opt}} - E_{CT}$. This implicitly assumes that the excited state stays on the same side of the PHJs and the energy transfer between the donor and the acceptor is negligible. However, as we show in the PL results, energy transfer was observed in some of the non-fullerene acceptor/polymer systems. The exact estimation of the efficiency of this energy transfer in the PHJ devices was difficult. Therefore, we assumed the opposite extreme case, that the excited energy is transferred completely to the material with the lower E_g^{opt} , which we assumed in the previous version of the manuscript. We plotted η_{gen} against the smaller $E_g^{\text{opt}} - E_{CT}$ of the donor and the acceptor. The results are shown in Fig. S13. Even with this assumption, the clear correlation between η_{gen} and $E_g^{\text{opt}} - E_{CT}$ remained unchanged.

The discussion above was added to the “Relationship between the state energy difference and charge generation efficiency” section. We have also added the excitation wavelengths to PL measurements in Fig. 6.

Reviewer #3 (Remarks to the Author):

The authors study the internal quantum yield and field-dependence of charge generation in 16 combinations of polymers with fullerene and non-fullerene acceptors in planar organic solar cells. They relate this to the driving force (difference of S1 energy to CT state energy or CS state energy) and find that high field-independent IQEs require a S1-CT difference of more than 0.2-0.3 eV, while the charge separation driving force (CT-CS) is not relevant. I-V characterization at different temperatures and light intensities, absorption/emission spectroscopy and UPS/LEIPS are carried out to determine the experimental parameters. The study is indeed very interesting to the OPV and electron transfer communities and could in principle be published in Nature Communications. There are however a number of conclusions that are currently not well enough substantiated, so that the manuscript needs major revisions before being re-considered for publication:

- The authors determine the energy levels by photoelectron, absorption and emission spectroscopy on the neat films and assume that they are the same in the PHJ devices. It is true that no intermixing or complicated morphologies occur in the PHJ structure, but the energy levels could still be affected by donor-acceptor interactions and electric fields at the interface. Could the authors comment on this?

First, we thank the reviewer for the insightful comments.

We agree with this comment. The problem is related to the definition of E_{CS} . In this study, the E_{CS} of the PHJ systems was defined as the difference between E_{HOMO}^D and E_{LUMO}^A for each combination. Although this definition is good as a first approximation, strictly speaking, the energy of the charge-separated state in OSCs is not independently determined by the energy levels of the materials and could be affected by the cell structures, such as the electrodes and film thickness. A good example can be seen in the V_{OC} dependence on the film thickness observed for PHJ OCSs recently reported by Izawa et al. (J. Phys. Chem. Lett. 2018, 9, 11, 2914-2918). Even in the same cell, the energy could depend on the positions of the charges considered. Thus, the material energy levels do not necessarily determine the cell properties, such as V_{OC} , completely, even for the very simple PHJs. Still, we think it is meaningful to show the relationship between the cell properties and E_{CS} deduced from E_{HOMO}^D and E_{LUMO}^A because they can be obtained by UPS and IPES measurements of the materials. We have clarified the definition of E_{CS} in this study according to this discussion.

- All driving forces reported in the manuscript are related to enthalpy, while it is actually the free energy (enthalpy and entropy) that determines if the processes are spontaneous. Do the authors think that entropy could play a role in their PHJ samples?

The clear correlation between η_{gen} and $E_{\text{g}}^{\text{opt}} - E_{\text{CT}}$ we found suggests that the charge transfer process from the S_1 and CT states can be described with Marcus theory and that this process strongly affects the overall charge generation process in organic solar cells. As the reviewer points out, it is the free energy of the system that should be the parameter for the analysis. However, the entropic gain for the transition from the S_1 to CT states is expected to be small because the interfacial CT states should be located close to the original S_1 state, which results in the small number of possible cases for the final state.

However, the exact mechanism of the efficient CT state splitting is still unclear. One explanation has been proposed based on a kinetic Monte Carlo simulation, that the Coulombic binding of the CT states can be overcome by the relaxation of electrons (holes) in the disordered density of states (DOS) of the acceptor (donor). Another explanation is that the entropic gain plays an important role in the CT states splitting into CS states. Either way, we conclude that the energetic (enthalpic) driving force is important for the S_1 to CT transition, whereas the relaxation in the disordered DOS or the entropic contribution may be more important for CT states splitting into CS states.

This discussion was included in the “Relationship between the state energy difference and charge generation efficiency” section.

- The CT state energy is obtained here by the temperature-dependent IV characteristics. Very often, sensitive EQE measurements (and EL) are used instead. Could the authors comment why they chose their method and if it is better?

According to the reviewer’s comments, we conducted highly sensitive EQE and EL measurements to evaluate the absorption and emission of the CT states in all the systems. For four PHJ systems out of the 16, we observed well-resolved CT bands and obtained E_{CT} values using the optical measurements with reasonable fittings. However, six systems had lower goodness of fit due to the deformation of the peak shapes, and the other six systems had poorly resolved E_{CT} bands owing to the S_1 bands and it was impossible to extract the information about the CT states. The extracted values are summarized in Table S3.

The E_{CT} values determined from the $qV_{\text{OC}}-T$ plots and EQE/EL measurements matched well for eight systems with deviations below 15% (Fig. S9 and Table S3), whereas two PHJ systems with the low fitting reliability showed larger deviations (47% and 30% for

BTA3/P3HT and PCBM/P3HT, respectively). We think that the accuracy of the EQE/EL technique may be limited in PHJs because the small D/A interface area leads to a smaller signal from interfacial absorption/emission relative to those from the bulk, which reduced the reliability of the gaussian fittings. Because of the lower reliability of the E_{CT} values determined by the EQE/EL measurements, we decided to use the E_{CT} values from the $qV_{OC}-T$ plots primarily for discussion. We confirmed that even when we used the E_{CT} values obtained by EQE/EL measurements, our discussion and conclusion remained unchanged (Fig. S12).

This discussion has been included in the revised manuscript (“Evaluation of E_{CT} ” section).

- The authors always use the S1 energy of the low bandgap component to define the S1-CT driving force (except PCBM). But then the IQE is spectrally integrated (i.e. taken from the JSC, solar spectrum and absorptance) and therefore refers to the photocurrent generated by both the electron and hole transfer processes, which have two different driving forces. It would be much more correct to obtain the IQE in a wavelength-resolved way and to relate the smaller driving force only to IQE at wavelengths where the low band gap component selectively absorbs.

According to the reviewer’s suggestion, we performed transfer matrix optical simulations for all the PHJ devices to calculate the absorptance of the organic layers without the parasitic absorption of Ag and ITO/glass substrate. By using this information, we reproduced the external quantum efficiency spectra of the devices. Based on an exciton diffusion model with an appropriate exciton collection length near the D/A interface, the charge generation efficiencies at the interface from the S_1 to CT states were evaluated separately for the donor and the acceptor. This analysis doubled the number of data points from 16 to 32 in the key figure (Fig. 4 in the revised manuscript). The charge generation efficiency decreased drastically as $E_g^{opt} - E_{CT}$ decreased.

In the above analysis, we used different E_g^{opt} values for the donors and the acceptors to calculate $E_g^{opt} - E_{CT}$. This implicitly assumes that the excited state stays on the same side of the PHJs and the energy transfer between the donor and the acceptor is negligible. However, as we show in the PL results, the energy transfer was observed in some of the non-fullerene acceptor/polymer systems. The exact estimation of the efficiency of this energy transfer in the PHJ devices was difficult. Therefore, we assumed the opposite extreme case, that the excited energy is transferred completely to the material with the lower E_g^{opt} , which we assumed in the previous version of the manuscript. We plotted η_{gen} against the smaller $E_g^{opt} - E_{CT}$ of the donor and the acceptor. The results are shown in Fig. S13. Even with this assumption, the clear

correlation between η_{gen} and $E_{\text{g}}^{\text{opt}} - E_{\text{CT}}$ remained unchanged. There was a decrease in the efficiency at larger $E_{\text{g}}^{\text{opt}} - E_{\text{CT}}$ of >0.8 eV, which is explained by Marcus charge transfer theory.

The discussion above was added to the “Evaluation of interfacial charge generation efficiency” and “Relationship between the state energy difference and charge generation efficiency” sections.

- Measuring IQE is a tricky task, usually requiring complex transfer matrix modelling of the photon densities inside the device. Can the authors comment on whether obtaining the absorptance in an integrating sphere is enough to get reliable results?

Please refer to the response to the comment above. The high reliability of the model is demonstrated by the matching of the calculated and observed absorption spectra and EQE spectra.

- For the electric field dependence, it is not clear how the “ratio of charge density” was obtained. Is it the ratio of IQEs?

The previous definition of the electric field dependence may be confusing. We have defined the electric field dependence in the revised manuscript as the ratio of the charge generation efficiency at an applied bias of 0 V (i.e., short circuit) to that at -1.0 V.

$$\text{Electric field dependence} = 100 - \frac{\eta_{\text{gen}} \text{ at } 0 \text{ V}}{\eta_{\text{gen}} \text{ at } -1.0 \text{ V}} \times 100 (\%)$$

To evaluate η_{gen} at a bias of -1.0 V, we conducted EQE measurements at a bias of -1.0 V and reproduced the spectra by using the simulated light absorptance in the same way for η_{gen} under short-circuit conditions. The lock-in technique was used for the biased EQE measurements to eliminate the dark current. Figures 5a and b show the electric field dependence plotted against $E_{\text{g}}^{\text{opt}} - E_{\text{CT}}$ and $E_{\text{CS}} - E_{\text{CT}}$, respectively. No clear trend was observed in the $E_{\text{CS}} - E_{\text{CT}}$ plot. In contrast, the $E_{\text{g}}^{\text{opt}} - E_{\text{CT}}$ plot showed that the charge generation become less dependent on the electric field with $E_{\text{g}}^{\text{opt}} - E_{\text{CT}}$ of around 0.4 eV. This conclusion is the same as in the previous version.

We have included this discussion in the “Electric field dependence of charge separation” section in the revised manuscript.

- For the bias-dependent emission, it would be important to know more precisely which of the donor or acceptor is emitting, and to control which one was excited. Now this is left open. If

the emission is always from the low bandgap component (also if the high bandgap one is excited), the S₁ state is formed by CT back-recombination. In this case, the field impacts the CT state. If the emission is always from the excited component, it means that the field impacts the charge transfer from the S₁ state.

We have added the PL excitation wavelength and which component (D or A) was excited to the PL measurements (Figs. 6, S14, and S15). In BTA3/J61 (Fig. 6a), the PL emission mainly came from the lower band-gap component (BTA3), although the donor (J61) was excited. Initially, we thought that this could be the evidence that the S₁ state is formed by CT back-recombination (as the reviewer suggested). However, we noticed that direct energy transfer from the donor to the acceptor prior to the emission is also possible. Currently, we cannot distinguish which process was dominant in the emission process. Because the energy transfer from the donor to the acceptor (or otherwise) at the interface would not be affected by the electric field, the observed change in PL intensity may be evidence for the electric field dependence of the S₁ state dissociation from either the donor or the acceptor. We have commented on this point in the “Electric field dependence of emission from the S₁ state” section.

Reviewers' comments:

Reviewer #1 (Remarks to the Author):

I agree with the revisions made, except with the one concerning the possible role of the triplet states.

In their rebuttal the authors mention the decay of the T1 state to the CT state. However the process more detrimental to the efficiency of the solar cell, is the decay of the CT state to the T1 state of either donor or acceptor. In donor-acceptor heterojunctions with a very large Egopt-ECT offset, the decay process from CT state to triplet state is inhibited if the CT state is lower in energy than the lowest triplet state. If the energy difference Egopt-ECT is made smaller, at a certain point the CT state will rise above the lowest lying triplet level and then recombination of the CT state to the lower lying triplet may become a major loss channel. Can the authors comment on this possibility?

Stefan Meskers

Reviewer #2 (Remarks to the Author):

The authors have satisfactorily addressed most of my comments. However, the description pertaining to the mechanism behind the "0.3 eV rule" is still lacking and should be addressed in more detail before publication in Nature Communications. This is in part because the main argument used in the present manuscript "interfacial disorder aiding charge separation" has been suggested before (e.g. DOI: 10.1021/ja505463r, DOI: 10.1002/adma.201304241), although not necessarily as the mechanism governing the "0.3 eV rule". If that is the authors claim, I think it should be substantiated. Due to this reason, and despite the fact that the author's study appears otherwise sound and thorough, I am on a split mind whether the conceptual novelty is sufficient for publication in Nature Communications. If the other referees review the manuscript favorably, I would also recommend publication, but would then advise the authors to expand their description on the following issues:

- The authors mention that Marcus theory is suitable to explain their charge generation efficiency versus EOPT – ECT results (e.g. particularly the inverted region). If this is the case, can the authors fit their Fig. 4b results using the Marcus charge transfer rate expressions? What does one learn from such a fit? Does the black line in the graphical abstract represent a Marcus theory fit? If possible, I think such a fit with an accompanying explanation would benefit the manuscript quite significantly, since it would point out the mechanism for the "0.3 eV rule" and a direction for future work.
- The authors suggest the presence of a disordered interfacial layer (e.g. disordered donor) between the donor and acceptor. If I understand the schematic in Fig. S17 correctly (by the way, SI contains a typo since it says Fig. 17 without "S"), wouldn't in the presence of such a disorder layer the charge-transfer photoluminescence (CT-PL) and charge-transfer electroluminescence (CT-EL) spectra be situated at different energy? I would expect the CT-PL spectra to be redshifted compared to the CT-EL spectra, since CT-PL originates from CT pairs directly at the D/A interface, whereas in CT-EL the injected charges would have to traverse the bulk donor and likely emit from lower energy CT states, as has been suggested earlier e.g. DOI: 10.1021/jp107587h.

Reviewer #3 (Remarks to the Author):

In response to the referee reports, the authors have carefully revised their manuscript and included major revisions. I find that they have responded very well to my comments as well as the ones of the other referees. The conclusions are now much better substantiated, and I support the paper for

publication in Nature Communications, since it appeals to a broad organic electronic and solar cell readership and describes important new insights. In particular, I like in the revisions that:

- They also attempted to get CT energy from EQE/EL measurements
- They did optical modelling
- They separated the charge generation efficiency from the donor and acceptor
- They show better PL data that reveals energy transfer

Very minor points to address (no new review necessary):

- Update record PCE, it is not 13% anymore
- p.3 | 45: rephrase that enthalpy is the driving force (free energy should be the driving force).

Point-by-point responses to reviewers' comments (the original comments are in *italic*):

Reviewers' comments:

Reviewer #1 (Remarks to the Author):

I agree with the revisions made, except with the one concerning the possible role of the triplet states.

In their rebuttal the authors mention the decay of the T1 state to the CT state. However the process more detrimental to the efficiency of the solar cell, is the decay of the CT state to the T1 state of either donor or acceptor. In donor-acceptor heterojunctions with a very large Egopt-ECT offset, the decay process from CT state to triplet state is inhibited if the CT state is lower in energy than the lowest triplet state. If the energy difference Egopt-ECT is made smaller, at a certain point the CT state will rise above the lowest lying triplet level and then recombination of the CT state to the lower lying triplet may become a major loss channel. Can the authors comment on this possibility ?

Stefan Meskers

We thank Prof. Meskers for giving us very insightful comments. Here we discuss the possible involvement of the triplet states in the loss process in more details. There are two possible situations:

(1) Through non-geminate recombination of electron and hole with the same spin direction. In this case, triplet CT state (^3CT) can be formed with the probability of 75% and then further relax into T_1 state (DOI: 10.1038/nature12339). As mentioned in the literature, this process can be a major loss channel in any type of organic photovoltaics. However, in our manuscript, we mainly discussed the charge generation process under short-circuit or reverse biased condition, where the effect of non-geminate recombination was shown to be negligible. Thus, we ignore the formation of ^3CT and T_1 states via non-geminate recombination in the charge generation process.

(2) Through geminate recombination. In this case, singlet ^1CT initially forms from S_1 , and then changes into ^3CT via intersystem crossing, and ^3CT relaxes into T_1 . Energetically, ^3CT to T_1 transition is possible depending on the relative alignment of T_1 state energies of the components and CT state energy. For example, the energy of T_1 is reported as 1 eV for PTB7 (DOI: 10.1038/srep29158) which are lower than E_{CT} in all the combination in the manuscript. On the other hand, some of the systems with P3HT showed higher T_1 energy than E_{CT} (E_{CT}

are in the range of 0.94-1.65 eV and the energy of T_1 was estimated as 1.4 eV for P3HT). Although we cannot completely deny the possible loss from T_1 state in this study, it is not certainly the dominant loss process because we observed the clear general correlations between $E_g^{\text{opt}} - E_{\text{CT}}$ and charge generation efficiencies. In addition, without strong spin-orbit coupling, the intersystem crossing from ^1CT to ^3CT is expected to be a slow process and at the same time, the typical lifetime of ^1CT for splitting to the free charges is fast (~ 1 ns) (DOI: 10.1126/science.1217745). Thus, we think ^1CT state is likely to be dissociated into free charges rather than to form ^3CT state. There have been several studies showing that the loss from geminately formed ^3CT state is not a major factor (DOI:10.1038/nature12339, 10.1038/srep29158). The effects of this loss process can also strongly depend on the kinetics and the local morphology. The reviewer reported that the T_1 state formation is suppressed in the BHJs with highly pure, well-defined D-A domain (DOI: 10.1002/adma.201001452). This is because ^1CT state splitting into CS state is efficient due to the local high carrier mobility and lowered effective energy of the charge separated state. In our PHJs, D and A are spatially well-separated, and the purity of each layer is close to 100%; hence, we think this situation is favorable for the charge separation from ^1CT but less favorable for the loss through ^1CT - ^3CT - T_1 transition. The quantitative analysis on the loss process from ^3CT would require the determinations of T_1 state energy for each component and the measurements on the current generation efficiency in the magnetic field (10.1063/1.4865203), which is however beyond the scope of this work.

Reviewer #2 (Remarks to the Author):

The authors have satisfactorily addressed most of my comments. However, the description pertaining to the mechanism behind the “0.3 eV rule” is still lacking and should be addressed in more detail before publication in Nature Communications. This is in part because the main argument used in the present manuscript “interfacial disorder aiding charge separation” has been suggested before (e.g. DOI: 10.1021/ja505463r, DOI: 10.1002/adma.201304241), although not necessarily as the mechanism governing the “0.3 eV rule”. If that is the authors claim, I think it should be substantiated.

We thank the reviewer for giving us precious comments.

The main argument of this work is the clear correlation between the charge generation efficiency and $E_g^{\text{opt}} - E_{\text{CT}}$ energetic difference. As we discussed in the manuscript, this correlation probably comes from Marcus charge transfer theory, which could be applicable regardless of the device architectures (PHJ or BHJ). However, it was reported that some BHJ systems exhibit efficient charge generation with negligible $E_g^{\text{opt}} - E_{\text{CT}}$. To explain this discrepancy, we hypothesized the possibility of the underestimation of $E_g^{\text{opt}} - E_{\text{CT}}$ value in BHJs due to the interfacial disordered layer. We still do not have any experimental data to prove this concept, but we believe it is meaningful for the OPV research community to point out the possible error of the estimation for the energetic driving force in BHJs. We plan to do experiments to prove the effects of the interfacial disorder by using the same PHJ setup in near future. We have cited the suggested manuscripts on the interfacial disorder in the revised version and revised the sentence on page 26 to make it clearer that the “interfacial disorder aiding charge separation” is a hypothesis to address the discrepancy between PHJ and BHJ.

Due to this reason, and despite the fact that the author’s study appears otherwise sound and thorough, I am on a split mind whether the conceptual novelty is sufficient for publication in Nature Communications. If the other referees review the manuscript favorably, I would also recommend publication, but would then advise the authors to expand their description on the following issues:

- *The authors mention that Marcus theory is suitable to explain their charge generation efficiency versus $E_{\text{OPT}} - E_{\text{CT}}$ results (e.g. particularly the inverted region). If this is the case, can the authors fit their Fig. 4b results using the Marcus charge transfer rate expressions? What does one learn from such a fit? Does the black line in the graphical abstract represent a Marcus theory fit? If possible, I think such a fit with an accompanying explanation would*

benefit the manuscript quite significantly, since it would point out the mechanism for the “0.3 eV rule” and a direction for future work.

In the previous manuscript, the black line in the graphical abstract was just added as a guide for eye, not representing the “parabola” of Marcus theory. As the reviewer’s comment, we tried to fit these data set using one parabola; however, we found the deviations were too large to extract any meaningful information from the theoretical fit. This may be due to the variations of the reorganization energy among the systems depending on the structures of the materials (DOI: 10.1021/jacs.6b12857). Instead, we plotted the one single parabola with assuming reorganization energy 0.3 eV in Fig. 4b. The upper limit of the charge generation efficiency well follows the parabola line in the Marcus normal region, supporting the validity of our analysis based on the theory. We have added the discussions above on page 18. We have also updated the image of the table of contents.

In principle, reducing the reorganization energy of the charge transfer event is the direct way to reduce the energetic requirement for the charge generation. To our knowledge, however, there is a fundamental limitation of the reduction of the reorganization energy of organic semiconductors originating from their carbon-carbon conjugated bonds (DOI: 10.1038/nenergy.2017.53). Since the reorganization energy could largely depend on the molecular structures, further generalization would be difficult from the viewpoint of the simple energetics. We should probably search for the special systems with low reorganization energy and analyze them in detail.

- The authors suggest the presence of a disordered interfacial layer (e.g. disordered donor) between the donor and acceptor. If I understand the schematic in Fig. S17 correctly (by the way, SI contains a typo since it says Fig. 17 without “S”), wouldn’t in the presence of such a disorder layer the charge-transfer photoluminescence (CT-PL) and charge-transfer electroluminescence (CT-EL) spectra be situated at different energy? I would expect the CT-PL spectra to be redshifted compared to the CT-EL spectra, since CT-PL originates from CT pairs directly at the D/A interface, whereas in CT-EL the injected charges would have to traverse the bulk donor and likely emit from lower energy CT states, as has been suggested earlier e.g. DOI: 10.1021/jp107587h.

We think the comment above “*I would expect the CT-PL spectra to be redshifted compared to the CT-EL spectra*” should read “*the CT-EL spectra to be redshifted compared to the CT-PL spectra*”. We agree with this expectation of the reviewer. The PL signal from CT state in BHJs can have higher energy than that of the EL as observed in the literature. We think we

need to be careful to correlate the PL signal with the charge generation, because the PL signal mainly comes from the CT states that have high probability of the recombination, while the CT states with high probability of dissociation can be silent in the PL measurement. The PHJs with intentionally inserting the very thin disordered layer between the D/A interface (*e.g.* D/disordered D/A; tri-layer structure) might be a good model to investigate these points. These experiments will be performed in future investigation. We have corrected the typo in SI as pointed out.

Reviewer #3 (Remarks to the Author):

In response to the referee reports, the authors have carefully revised their manuscript and included major revisions. I find that they have responded very well to my comments as well as the ones of the other referees. The conclusions are now much better substantiated, and I support the paper for publication in Nature Communications, since it appeals to a broad organic electronic and solar cell readership and describes important new insights. In particular, I like in the revisions that:

- *They also attempted to get CT energy from EQE/EL measurements*
- *They did optical modelling*
- *They separated the charge generation efficiency from the donor and acceptor*
- *They show better PL data that reveals energy transfer*

We thank the reviewer for the very positive comments.

Very minor points to address (no new review necessary):

- *Update record PCE, it is not 13% anymore*
- *p.3 l 45: rephrase that enthalpy is the driving force (free energy should be the driving force).*

We updated the record PCE value of OPV to 15% by citing 10.1016/j.joule.2019.01.004. We have rephrased the term to “energetic driving force” to avoid the confusion with the free energy.

REVIEWERS' COMMENTS:

Reviewer #1 (Remarks to the Author):

Nakano and coworkers report an interesting correlation between charge generation efficiency and the energy difference between S1 excited state and the charge transfer state ($E_{g,opt-ECT}$). The charge generation efficiency seems to reach its maximum when the above mentioned energy difference reaches 0.2-0.3 eV. There seems to be no clear correlation between charge generation efficiency and the energy difference between charge transfer state and charge separated state. These are in my is a very meaningful experimental insights.

I agree with reviewer 2 that the discussion by the authors on their '0.2-0.3 eV' rule provided by the authors is not satisfactory:

- The charge generation efficiency must be a competition between various processes each with its own rate and activation parameters. One of the processes in competition with charge generation could be formation of triplet excited states, but the authors chose to not mention this option at all apparently within any backup of experimental data.
- The authors provide the reader with a hypothesis that in bulk heterojunctions $E_{g,opt} - ECT$ is 'underestimated'.

Overall, in my view the authors provide a highly simplified analysis of their findings and seem to lack further experimental clues to arrive at a more detailed picture of what is actually going on during the charge generation process.

I'm willing to support publication of this after the authors have modified their discussion section pointing out that the charge generation is in fact a more complicated process than a single Marcus type transition between singlet excited state and charge transfer state.

Stefan Meskers

Reviewer #2 (Remarks to the Author):

The authors have satisfactorily addressed my comments and I recommended publication in Nature Communications.

For the final version, the authors should address (no review necessary) the following minor points:

- If I understand correctly, the grey line fit in Fig. 4b is a Gaussian fit following Marcus theory, and not a "parabola" fit. If so, the authors should correct the corresponding sentences - "parabola" should read "Gaussian".
- As a follow up comment, it would be useful if the authors would explicitly show/explain the equation used to fit the Fig. 4b data.

Armantas Melianas

Point-by-point responses to reviewers' comments (the original comments are in *italic*):

Reviewers' comments:

Reviewer #1 (Remarks to the Author):

Nakano and coworkers report an interesting correlation between charge generation efficiency and the energy difference between S1 excited state and the charge transfer state (Eg,opt-ECT). The charge generation efficiency seems to reach its maximum when the above mentioned energy difference reaches 0.2-0.3 eV. There seems to be no clear correlation between charge generation efficiency and the energy difference between charge transfer state and charge separated state. These are in my is a very meaningfull experimental insights.

I agree with reviewer 2 that the discussion by the authors on their '0.2-0.3 eV' rule provided by the authors is not satisfactory:

- The charge generation efficiency must be a competition between various processes each with its own rate and activation parameters. One of the processes in competition with charge generation could be formation of triplet excited states, but the authors chose to nit mention this option at all apparently within any backup of experimental data.*
- The authors provide the reader with a hypothesis that in bulk heterojunctions Eg,opt – ECT is 'underestimated'.*

Overall, in my view the authors provide a highly simplified analysis of their findings and seem to lack further experimental clues to arrive at a more detailed picture of what is actually going on during the charge generation process.

I'm willing to support publication of this after the authors have modified their discussion section pointing out that the charge generation is in fact a more complicated process that a single Marcus type transition between singlet excited state and charge transfer state.

Stefan Meskers

We thank Prof. Meskers for giving us very insightful comments through the peer-review process. We have modified the discussion part to point out the complexity of the charge

generation process of organic solar cells, which is possibly competing with triplet state formation (page 16, 2nd paragraph).

Reviewer #2 (Remarks to the Author):

The authors have satisfactorily addressed my comments and I recommended publication in Nature Communications.

For the final version, the authors should address (no review necessary) the following minor points:

- If I understand correctly, the grey line fit in Fig. 4b is a Gaussian fit following Marcus theory, and not a “parabola” fit. If so, the authors should correct the corresponding sentences*
- "parabola" should read "Gaussian".*
- As a follow up comment, it would be useful if the authors would explicitly show/explain the equation used to fit the Fig. 4b data.*

Armantas Melianas

We thank Dr. Melianas for giving us very insightful comments through the peer-review process. We changed the word “parabola” to “Gaussian” through the manuscript. We added the detail of the Gaussian plot in Supplementary Note 5.